


# Preparation of primary standard mixtures for atmospheric oxygen measurements with uncertainty less than 1 ppm for oxygen mole fractions

Nobuyuki Aoki[1], Shigeyuki Ishidoya[2], Nobuhiro Mastumoto[1], Takuro Watanabe[1], Takuya Shimosaka[1], and Shohei Murayama[2]

[1] National Meteorology Institute of Japan, National Institute of Advanced Industrial Science and Technology (AIST), Tsukuba, 305-8563, Japan

[2] National Institute of Advanced Industrial Science and Technology (AIST), Tsukuba, 305-8569, Japan

*Correspondence to*: Nobuyuki Aoki (aoki-nobu@aist.go.jp)

**Abstract.** Primary standard mixtures with less than 1 ppm or 5 per meg standard uncertainty for $O_2$ mole fractions or for $O_2/N_2$ ratios were prepared to monitor changes, which occurred in atmospheric oxygen. These mixtures were crafted in 10 L high-pressure aluminum cylinders using a gravimetric method in which unknown uncertainty factors were identified and subsequently reduced. The mole fractions of the constituents, $CO_2$, Ar, $O_2$, and $N_2$, were mainly determined using the masses of the respective source gases that had been filled into the cylinders. To precisely determine the masses of the source gases used in each case, the differences in the masses of the cylinders before and after filling were calculated and compared to nearly identical reference cylinders. Although the mass of the cylinder with respect to the reference cylinder tended to vary in relation to temperature differences between both cylinders, the degree of change could be reduced by measuring both cylinders at the same temperature. The standard uncertainty for the cylinder mass was determined to be 0.82 mg. The standard uncertainties for the $O_2$ mole fractions and $O_2/N_2$ ratios in the primary standard mixtures ranged from 0.7 ppm to 0.8 ppm and from 3.3 per meg to 4.0 per meg, respectively. Based on the primary standard mixtures, the mole fractions of atmospheric $O_2$ and Ar on Hateruma Island, Japan. In 2015, the $O_2$ and Ar mole fractions were found to be $209339.1 \pm 1.1$ ppm and $9334.4 \pm 0.7$ ppm.

## 1 Introduction

Observation of atmospheric $O_2$ mole fractions provides important information about the global carbon cycle (Keeling and Shertz, 1992; Bender et al., 1996; Keeling et al., 1996, 1998a; Stephens et al., 1998; Battle et al., 2000; Manning and Keeling, 2006). For example, long-term observation allows the estimation of land biotics and oceanic $CO_2$ uptake (Manning and Keeling, 2006; Tohjima et al., 2008; Ishidoya et al., 2012a, 2012b). Various measurement techniques have been developed for this purpose, including the utilization of interferometry (Keeling et al., 1998b), mass spectrometry (Bender et al., 1994; Ishidoya et al., 2003; Ishidoya and Murayama, 2014), a paramagnetic technique (Manning et al., 1999; Aoki et al., 2017; Ishidoya et al., 2017), a vacuum-ultraviolet absorption technique (Stephens et al., 2003), gas chromatography (Tohjima, 2000), and a method that uses fuel cells (Stephens et al., 2007; Goto et al., 2013). In all of these cases, the calibration using standard mixtures is required to precisely determine the relationship between the analyzers' outputs and $O_2$ mole fraction values obtained.

The mole fraction of atmospheric $O_2$ is commonly expressed as a function of $O_2/N_2$ ratio relative to an arbitrary reference (Keeling and Shertz, 1992), according to Eq. (1).



$$\delta(O_2/N_2)(\text{per meg}) \ = \ [\frac{(O_2/N_2)_{\text{sample}}}{(O_2/N_2)_{\text{standard}}} - 1] \times 10^6 \qquad\qquad (1)$$

In this equation, the subscripts "sample" and "standard" refer to a sample air and a standard air, respectively. As the

$O_2$ mole fraction of air is 20.946 %, a change of 4.8 per meg in $\delta(O_2/N_2)$ corresponds to a change of 1 $\mu$mol mol$^{-1}$ in

the $O_2$ mole fraction. In this study, the unit of "$\mu$mol mol$^{-1}$" is abbreviated as "ppm."

There are approved primary standard mixtures for use in these types of experiments for $CO_2$, $CH_4$, and $N_2O$, which

are prepared using either manometry (Zhao et al., 1997) or gravimetry (Tanaka et al., 1983; Matsueda et al., 2004;

Dlugokencky et al., 2005; Hall et al., 2007). Tohjima et al. (2005) first prepared primary standard mixtures for

observation of atmospheric $O_2$ using a gravimetric method in which the standard uncertainties were noted at 15.5 per

meg for the $O_2/N_2$ ratio and 2.9 ppm for the $O_2$ mole fraction. Since the 2.9 ppm standard uncertainty recorded by

Tohjima et al. was much larger than the gravimetrically expected value of 1.6 ppm, it was suggested that there are

unknown factors exerting influence on the mass readings of the cylinders.

Reported peak-to-peak amplitudes of seasonal cycles and trends for atmospheric $\delta(O_2/N_2)$ were within the range of 50

per meg to 150 per meg (from 10 ppm to 30 ppm for $O_2$ mole fractions) and $-20$ per meg yr$^{-1}$ ($-4$ ppm yr$^{-1}$ for $O_2$

mole fractions), respectively (Keeling et al., 1993; Battle et al., 2000; Van der Laan–Luijkx et al., 2013). To monitor

these slight variations, it was recommended to develop primary standard mixtures with $O_2/N_2$ ratios that had standard

uncertainty of less than 5 per meg or $O_2$ mole fractions that had standard uncertainty of less than 1 ppm (Keeling et

al., 1993; WMO, 2016). In this study, primary $O_2$ standard mixtures with the recommended uncertainty of less than 5

per meg or 1 ppm is hereafter expressed as "a highly precise $O_2$ standard mixture."

Since the variations in atmospheric $O_2$ were less than 500 per meg (100 ppm) (Bender et al., 1994; Tohjima, 2000;

Stephens et al., 2007; Goto et al., 2013), the highly precise $O_2$ standard mixtures used to monitor atmospheric $O_2$

required the use of a range of 500 per meg (100 ppm) upwards. The resultant standard uncertainty would be higher

than the recommended uncertainty, which could interfere with its corresponding slope of calibration line in an analyzer

used for the monitoring. For example, when two standard gases that had uncertainty values of 3 ppm (15 per meg) and

the difference in both $O_2$ mole fractions of 100 ppm (500 per meg) were used for calibration of an analyzer, the slope

of the calibration line calculated for the analyzer would reflect a 6 % deviation from the actual value if one cylinder

would have $O_2$ mole fraction which would be 3 ppm higher than the true level while the other cylinder would have a

deviation that was 3 ppm lower than the true level. Given this, it is important to verify not only the scale but also its

corresponding slope for each laboratory's standard gas mixtures using highly precise $O_2$ standard mixtures. Because

the highly precise $O_2$ standard mixtures have not been yet developed, there has been a need for their development.

Our laboratory has built upon a weighing system proposed by Matsumoto et al. (2004) in which gravimetry was used

to prepare standard mixtures. This system allows accurate weight measurements in which the standard uncertainty is

2.6 mg. The integration of a new mass comparator with better repeatability have been made to the weighing system.

In this study, we developed a means of identifying and minimizing unknown uncertainty factors that contributed to

deviations in the mass readings of the cylinders during preparation of the highly precise $O_2$ standard mixtures with the

weighing system. The standard uncertainties for the mole fractions of various constituents in the highly precise $O_2$



standard mixtures, which have been prepared using this improved weighing means, are discussed. Additionally, the constituents in the standard mixtures was validated by measuring the mole fractions of $CO_2$ and $O_2$, as well as both $Ar/N_2$ and $O_2/N_2$ ratios. To validate the scale of $O_2/N_2$ ratio at National Institute of Advanced Industrial Science and Technology (AIST) determined using the highly precise $O_2$ standard mixtures, the $O_2/N_2$ ratios for air samples

collected at Hateruma Island, Japan obtained from our measurements were preliminary compared with the $O_2/N_2$ ratios at Hateruma Island on the scale of National Institute for Environmental Studies (NIES) determined by Tohjima et al. (2005). Also, the mole fractions for Ar and $O_2$ in air samples at Hateruma Island were determined and compared with previously reported values.

## 2 Materials and Methods

**2.1 Weighing procedure for a high-pressure cylinder**

The highly precise $O_2$ standard mixtures were prepared in 10 L aluminum cylinders (Luxfer Gas Cylinders, UK), which had a diaphragm valve (G-55, Hamai Industries Limited, Japan) with poly(chlorotrifluoroethylene) (PCTFE) as sealant. The cylinder filled with highly precise $O_2$ standard mixture was hereafter referred to as "gravimetric cylinder." The masses obtained for the gravimetric cylinders were determined using a weighing system which is the

same as that reported by Matsumoto et al (2004) except a mass comparator. The mass comparator used in the research of Matsumoto et al. was replaced with a new mass comparator (XP26003L, Mettler Toledo, Switzerland), which had a maximum capacity of 26.1 kg, a readability of 1 mg, and a linearity of 20 mg. The mass measurements for the gravimetric cylinders were performed in a weighing room in which temperature and humidity were controlled at $26 \pm 0.5$ ℃ and $48 \pm 1$ %, respectively. The temperature, humidity, and atmospheric pressure surrounding the weighing

system were measured using a USB connectable logger (TR-73, T & D Corporation, Japan).

The mass measurement of each gravimetric cylinder was conducted with respect to a nearly identical reference cylinder aiming to reduce any influence exerted by zero-point drifts, sensitivity issue associated with the mass comparator, changes in buoyancy acting on the cylinder, and/or adsorption effects on the cylinder's surface as a result of the presence of water vapor (Alink et al., 2000; Milton et al., 2011). Each weighing cycle for both the gravimetric

and reference cylinders consisted of several consecutive weighing operations in the ABBA order sequence, where "A" and "B" denote the reference and gravimetric cylinder, respectively. The process of loading and unloading the cylinders was automated. One complete cycle of the ABBA sequence required five minutes. The mass reading recorded from the weighing system was given by the mass difference, which was computed by subtracting the reference cylinder reading from the gravimetric cylinder reading.

Generally, the outputs of mass comparators are known to be nonlinear, as such, there is a tendency to underestimate or to overestimate the differences in the mass values obtained after each reading. This is because the calibration lines of the comparator tend to be different for various scale ranges. To reduce the influence of this nonlinearity, the cylinders were weighed only when the weight difference between the gravimetric and reference cylinders was less than 500 mg. This was achieved by placing standard weights in the weighing pan alongside each cylinder. Any mass

differences obtained for our weighing system took into account the masses and the buoyancies of the standard weights. The masses of the standard weights were traced to International System of Units. The standard uncertainties of the



masses were 0.25 mg, 0.045 mg, 0.028 mg, 0.022 mg, 0.018 mg, 0.014 mg, 0.011 mg, and 0.0090 mg for the 500 g, 100 g, 50 g, 20 g, 10 g, 5 g, 2 g, and 1 g, respectively.

## 2.2 Preparation of the highly precise $O_2$ standard mixtures

Eleven highly precise $O_2$ standard mixtures were prepared in accordance with ISO 6142-1:2015. Pure $CO_2$ (>99.998 %,

Nippon Ekitan Corporation, Japan), pure Ar (G1-Grade, 99.9999 %, Japan Fine Products, Japan), pure $O_2$ (G1-Grade, 99.9999 %, Japan Fine Products, Japan), and pure $N_2$ (G1-Grade, 99.9999 %, Japan Fine Products, Japan) were used as soruce gases. The value of $\delta^{13}C$ in pure $CO_2$ (which was adjusted to the atmospheric level) was −8.92 ‰ relative to Vienna Pee Dee Belemnite (VPDB). Impurities in the source gases were identified and quantified using a gas chromatograph with a thermal conductivity detector for $N_2$, $O_2$, $CH_4$ and $H_2$ in pure $CO_2$, a gas chromatograph with a

mass spectrometer for $O_2$ and Ar in pure $N_2$ and $N_2$ in pure $O_2$, a Fourier transform infrared spectrometer for $CO_2$, $CH_4$ and CO in pure $N_2$, $O_2$, and Ar, a galvanic cell-type $O_2$ analyzer for $O_2$ in pure Ar, a capacitance-type moisture meter for $H_2O$ in pure $CO_2$, and a cavity ring-down-type moisture meter for $H_2O$ in pure $N_2$, $O_2$ and Ar.

First, standard mixtures of $CO_2$ in Ar were prepared from pure $CO_2$ and pure Ar using the gravimetric method. The molar ratios of $CO_2$ to Ar were close to the atmospheric ratio of Ar (9340 ppm) to $CO_2$ (400 ppm or 420 ppm). Next,

the gravimetric cylinders were filled as follows with the mixtures of $CO_2$ in Ar, pure $O_2$ and pure $N_2$ in a filling room in which the temperature was controlled at 23 ± 1 ℃ and humidity was not controlled. The gravimetric cylinder was evacuated using a turbomolecular pump before being weighed using the ABBA technique. Afterward, the evacuated cylinder was filled with the $CO_2$ in Ar standard mixture and weighed again. The mass of the filled $CO_2$ in Ar standard mixture was determined by the difference in mass before and after filling. The masses of filled pure $O_2$ and $N_2$ were

also treated in the same manner. The final pressure in the cylinder was 12 MPa, and the masses of the individual gases were approximately 8 g of the $CO_2$ in Ar standard mixture, 300 g of pure $O_2$, and 1000 g of pure $N_2$.

## 2.3 Analytical methods

To validate the constituents in the highly precise $O_2$ standard mixtures, the constituents were measured using a cavity ring-down spectrometer for measuring the mole fraction of $CO_2$, a mass spectrometer for measuring the $Ar/N_2$ and

$O_2/N_2$ ratios, and a paramagnetic $O_2$ analyzer for measuring the mole fraction of $O_2$.

### 2.3.1 Measurement of $CO_2$ mole fraction

The mole fractions of $CO_2$ were measured using a cavity ring-down spectrometer (G2301, Picarro, USA), which was equipped with a multi-port valve (Valco Instruments Co. Inc., USA) for gas introduction and a mass flow controller (SEC-N112, 100SCCM, Horiba STEC, Japan). Mole fractions were determined using three primary standard gases

(364.50 ± 0.14 ppm, 494.04 ± 0.14 ppm, and 500.32 ± 0.14 ppm) that had been prepared from pure $CO_2$ and purified Air (G1 grade, Japan Fine Products, Japan) in accordance with ISO 6142-1:2015, respectively. The value of $\delta^{13}C$ in pure $CO_2$ (which was adjusted to the atmosphere level) was −8.92 ‰ relative to VPDB.

### 2.3.2 Measurement of $O_2/N_2$ and $Ar/N_2$ ratios

The $O_2/N_2$ and $Ar/N_2$ ratios were measured using a mass spectrometer (Thermo Scientific Delta-V) (Ishidoya and

Murayama, 2014). The $O_2/N_2$ ratio is expressed as $\delta(O_2/N_2)$ according to Eq. (1). The $Ar/N_2$ ratio, which is also expressed as $\delta(Ar/N_2)$, is defined by




$$\delta(Ar/N_2)(\text{per meg}) = \left[\frac{(Ar/N_2)_{sample}}{(Ar/N_2)_{standard}} - 1\right] \times 10^6 \qquad (2)$$

where the subscripts "sample" and "standard" refer to the sample air and standard air in the same way as $\delta(O_2/N_2)$,
respectively. In this study, natural air in 48 L aluminum cylinder (Cylinder No. CRC00045), equipped with a
5    diaphragm valve (G-55, Hamai Industries Limited, Japan) was used as the standard air to determine the $\delta(O_2/N_2)$ and
$\delta(Ar/N_2)$ values on the AIST scale (Ishidoya and Murayama, 2014). The mass spectrometer was adapted to
simultaneously measure ion beam currents for masses 28 ($^{14}N^{14}N$), 29 ($^{15}N^{14}N$), 32 ($^{16}O^{16}O$), 33 ($^{17}O^{16}O$), 34 ($^{18}O^{16}O$),
36 ($^{36}Ar$), 40 ($^{40}Ar$), and 44 ($^{12}C^{16}O^{16}O$). These masses were also noted as deviations in $\delta(^{15}N^{14}N/^{14}N^{14}N)$,
$\delta(^{17}O^{16}O/^{16}O^{16}O)$, $\delta(^{18}O^{16}O/^{16}O^{16}O)$, $\delta(^{16}O^{16}O/^{14}N^{14}N)$, $\delta(^{36}Ar/^{40}Ar)$, $\delta(^{40}Ar/^{14}N_2)$, and $\delta(^{12}C^{16}O^{16}O/^{14}N^{14}N)$ from the
corresponding atmospheric values that had been recorded for the standard air.

In the case of sample air, it was assumed that both the $\delta(O_2/N_2)$ and $\delta(Ar/N_2)$ values were equal to those of $\delta(^{16}O^{16}O$
$/^{14}N^{14}N)$ and $\delta(^{40}Ar/^{14}N^{14}N)$, since the ratios of Ar, O, and N isotopes present in the atmosphere tended to be
spatiotemporally constant. On the other hand, the isotopic ratios of pure Ar, $O_2$, and $N_2$ used in this study were different
from the atmospheric values listed in Table 1. Consequently, both the $\delta(O_2/N_2)$ and $\delta(Ar/N_2)$ values in the highly
precise $O_2$ standard mixtures were computed using the measurements obtained for $^{15}N^{14}N/^{14}N^{14}N$ , $^{17}O^{16}O/^{16}O^{16}O$ ,
$^{18}O^{16}O/^{16}O^{16}O$ , $^{36}Ar/^{40}Ar$, and $^{38}Ar/^{40}Ar$, as depicted in the equations below.

$$\delta(O_2/N_2) = \left\{ \frac{(^{16}O^{16}O/^{14}N_2)_{STD}}{(^{16}O_2/^{14}N_2)_{standard}} \times \right.$$

$$\left. \left[\frac{1+^{17}O^{16}O/^{16}O^{16}O+^{18}O^{16}O/^{16}O^{16}O}{1+^{15}N^{14}N/^{14}N^{14}N}\right]_{STD} \middle/ \left[\frac{1+^{17}O^{16}O/^{16}O^{16}O+^{18}O^{16}O/^{16}O^{16}O}{1+^{15}N^{14}N/^{14}N^{14}N}\right]_{standard} - 1\right\} \times 10^6$$

$$(3)$$

$$\delta(Ar/N_2) = \left\{ \frac{(^{40}Ar/^{14}N^{14}N)_{STD}}{(^{40}Ar/^{14}N^{14}N)_{standard}} \times \left[\frac{1+^{36}Ar/^{40}Ar+^{38}Ar/^{40}Ar}{1+^{15}N^{14}N/^{14}N^{14}N}\right]_{STD} \middle/ \left[\frac{1+^{36}Ar/^{40}Ar+^{38}Ar/^{40}Ar}{1+^{15}N^{14}N/^{14}N^{14}N}\right]_{standard} - 1\right\} \times 10^6$$

$$(4)$$

The subscripts "STD" refer to the highly precise $O_2$ standard mixtures that were prepared in this study. The values of
$^{15}N^{14}N/^{14}N^{14}N$, $^{17}O^{16}O/^{16}O^{16}O$, and $^{18}O^{16}O/^{16}O^{16}O$  in both the $O_2$ standard mixtures and standard air were calculated
using the isotope abundances of O and N listed in Table 1. The $^{36}Ar/^{40}Ar$ ratio for the highly precise $O_2$ standard
mixtures was calculated from $\delta(^{36}Ar/^{40}Ar)$ and $(^{36}Ar/^{40}Ar)_{standard}$. The value of $\delta(^{36}Ar/^{40}Ar)$ were determined using the
mass spectrometer. The $(^{36}Ar/^{40}Ar)_{standard}$  was determined using the atmospheric value ($^{36}Ar/^{40}Ar = 0.003349 \pm$
$0.000004$), because the ratio of Ar isotopes in standard air was equal to that of the atmospheric value. On the other
hand, the value of $^{38}Ar/^{40}Ar$ in the highly precise $O_2$ standard mixtures was $^{38}Ar/^{40}Ar = 0.000631 \pm 0.000004$ which
was atmospheric values. The atmospheric values of abundance for Ar isotopes were reported in an IUPAC technical
report (Böhlk, 2014).

**2.3.3 Measurement of O₂ mole fractions**



A paramagnetic oxygen analyzer (POM-6E, Air Liquide Japan) was used to measure the mole fractions of $O_2$ in the highly precise $O_2$ standard mixtures. Details regarding the analyzer used have been reported by Aoki and Shimosaka (2017). Briefly, the analyzer was equipped with inlets for sample and reference gases (Kocache, 1986). Synthetic air with $O_2$ mole fraction of 20.650 % was used as the reference gas, and the pressures of the reference gas and the sample
5 gas were set at 300 kPa, and 180 kPa, respectively.

## 3 Identifying and minimizing unknown factors of uncertainty

As mentioned before, there were several unknown factors that influenced the differences in mass obtained for the gravimetric and reference cylinders. These factors in uncertainty and the weighing procedure used to minimize them are discussed in this section.

10 Generally, the mass reading of a cylinder obtained from a mass comparator tends to vary as a result of numerous factors. Buoyancy effects can be caused by changes in the density of the surrounding air due to the variations in ambient temperature, humidity, and pressure, whereas adsorption effects can greatly influence mass readings of the cylinder by the adsorption and desorption of water vapor in surrounding ambient air on the external surface of the cylinder (Alink et al., 2000; Mizushima, 2004, 2007; Milton et al., 2011). Thermal effects are related to the temperature

15 gradients between the cylinder and surrounding ambient air (Gläser, 1990, 1999; Mana et al., 2002; Gläser and Borys, 2009; Schreiber et al., 2015). They change a weight force of the cylinder through friction forces exerted on the vertical surface of the cylinder and pressure forces on the horizontal surface. Both the friction and pressure forces are caused by the upward or downward flow of air, which was cooled or heated by the cylinder. Mass differences between the gravimetric and reference cylinders tend to deviate from true value when these effects are exerted independently and

20 to varying degrees on the gravimetric and reference cylinders.

When the ABBA technique is used to perform mass measurements, the deviations become negligible because they are equally exerted on both the gravimetric and reference cylinders under identical experimental conditions. Actually, any buoyancy effects could be canceled by adopting the ABBA technique in our mass measurements (see Section 4.3.1). However, the temperature on the gravimetric cylinder's surface could change by adiabatic compression of the source

25 gases and the work (evacuating and filling) in the filling room where is different from the weighing room in temperature, whereas adsorption water amounts on the gravimetric cylinder's surface could change by the work in the filling room where is different from the weighing room in humidity. This non-uniformity was assumed to be the main contributor of uncertainties in the obtained mass values (Matsumoto et al., 2008). Therefore, we examined achievement of the equilibrium in both humidity and temperature for the gravimetric cylinder's surface, as well as the

30 surrounding ambient air, before carrying out any measurement for identifying and minimizing the contribution of the non-uniformity.

### 3.1 The time required for equilibration with ambient air

Achieving temperature and humidity equilibrium between the cylinder's surface and surrounding ambient air could be done by placing the cylinder on the weighing system for an appropriate time interval before mass readings. Here

35 the equilibrium at the reference cylinders' surface always maintained because the reference cylinder had been left on the weighing system, whereas the equilibrium of the gravimetric cylinder's surface had often been disturbed by

processes of the cylinder evacuation and the gas filling. To quantify the time needed for equilibration after the disturbing, the mass differences between the gravimetric and reference cylinders recorded after evacuation of the gravimetric cylinder and subsequent filling of the source gases were monitored. The values were plotted against the time needed to achieve equilibrium (Figure 1). The equilibrium was considered to be achieved when the standard

deviation of the values remained constant for two or more hours and were less than the repeatability value of 0.82 mg (see in Section 4.3.1.). Interesting, the mass differences recorded after evacuating and filling with the $CO_2$ in Ar mixture tended to decrease as time elapsed while those after filling with pure $O_2$ and the $N_2$ gases tended to increase. The time needed for equilibration is defined as the time elapsed from cylinder evacuation or filling to the point of equilibrium. The equilibrium time was noted as 5 h after complete cylinder evacuation. The times needed to achieve

the equilibrium after the cylinders were filled with the relevant gas were different between the filled gas species to some extent. For the $CO_2$ in Ar mixture, the equilibrium was achieved in 3 h to 5 h while 4 h to 5 h were required for $O_2$ equilibration and 7 h to 9 h for $N_2$. It is considered that each equilibrium time have some connection with the temperature of the gravimetric cylinder just after the evacuation and the gas filling, since the mass readings of the gravimetric cylinder decreases depending on increase in its surface temperature as for either thermal effect or

adsorption effect. This is because the temperature differences between the gravimetric and reference cylinders was the main factor contributing to the friction and pressure forces of thermal effect at room temperature. The mass difference decreases as the temperature of the gravimetric cylinder becomes higher than that of the reference cylinder. On the other hand, amount of adsorbed water on gravimetric cylinder's surface also decreases with increase of its temperature. The mass difference decrease as the temperature of the gravimetric cylinder becomes higher than that of the reference

cylinder.

Actually, the deviations in the mass difference values shown in Figure 1 had some connection with the temperature of the gravimetric and reference cylinders, because the gravimetric cylinder's temperature recorded after the evacuation was 2 K lower while the temperatures recorded after filling with the standard $CO_2$ in Ar mixture, pure $O_2$, and pure $N_2$ were −0.7 K, 1 K, and 6 K higher, respectively, than that of the reference cylinder. On the other hand, the

temperature of the gravimetric cylinder after the evacuation and the filling depends on amounts of the source gases and the conditions of the weighing room. Considering this, a reference parameter to clearly identify when equilibrium had been achieved was needed to determine more accurately the mass differences between the cylinders and to minimize associated factors of uncertainty.

### 3.2 Deviation of the mass difference by thermal effect

The relationship between the deviation values obtained in the recorded mass differences and the temperature differences on the surface of the gravimetric and reference cylinders was investigated. The results of the closed squares shown in Figure 2 indicate that the deviation was proportional to the temperature differences and slope of the fitting line, which had been obtained by applying linear least square methods to the data. This deviation rate was determined to be −14.3 mg $K^{-1}$. Although the results indicate that a temperature difference of 0.1 K caused a deviation of 1.4 mg,

the deviation in the recorded mass differences ensures the repeatability value of 0.82 mg that is achieved by reducing the temperature difference to below 0.06 K. By conducting measurements of the cylinder temperatures using a thermocouple-type thermometer with the resolution of 0.1 K (TX1001 digital thermometer, probe-90030, Yokogawa





Test & Measurement Corporation, Tokyo, Japan) and ensuring that the readings were taken when the temperature of both cylinders indicated the same values, we were able to reduce the deviation contributing to the mass difference.

To validate the proposed weighing procedure, the reproducibility of the mass difference values obtained after disturbing the equilibrium had to be evaluated. Hence, the reading sequence after a cooling or heating cycle of the

cylinders was examined. Figure 3 illustrates the results in which four heating cycles (number 1 to 4) and four cooling cycles (number 5 to 8) were conducted. In this experiment, the temperatures of the cooled or heated cylinder were 1 K to 3 K lower or 10 K to 20 K higher, respectively, than that of the reference cylinder. When the masses were recorded after the temperatures of both the gravimetric and reference cylinders were equivalent, no difference in the values recorded after the cooling and heating cycles was noticed. The reproducibility of the mass difference values

was estimated to be 0.44 mg with regards to the standard deviation of the mass difference values shown in Figure 3. The fact that the standard deviation was lower than the repeatability values confirmed the validity of the weighing procedure and indicated that the changes in the mass differences attributable to non-equilibrium conditions were negligible. It was confirmed that the proposed weighing procedure had a repeatability of 0.82 mg.

It is difficult to state whether changes in the mass differences recorded for the cylinders was caused by thermal or

adsorption effects simply by analyzing these results. This is because both effects are related to temperature fluctuations. However, an important indication that the changes were caused by one factor or the other is related to the fact that thermal effects influenced the slope of the calibration line solely through temperature fluctuations, whereas the adsorption effects influenced the slope of the calibration line via a combination of both ambient temperature and humidity. This is due to the fact that the adsorbed or desorbed amounts of water on the surface of both cylinders is

highly dependent on the cylinders' temperature, humidity of the surrounding ambient air, and condition of the cylinder's surface. To determine which of these effects contributed the most to the changes in the mass readings, the relationship between the deviations and temperature differences was investigated under various conditions in the weighing room. Humidity was strictly controlled at 30 %, 50 %, 65 %, and 80 %, whereas the temperature levels were maintained at 22 ℃, 26 ℃, and 29 ℃. As shown in Figure 2, the results indicated that the deviation values did not

depend on the humidity and temperature factors. These results indicated that the dominant factor of changes in the mass difference values was temperature-related and not an effect of adsorption. Therefore, we focused on minimizing the impact of any thermal effects during the further experiments.

## 4 Preparation of the $O_2$ Standard Mixtures

In this section, we discuss any uncertainty factors associated with the mole fractions of the constituents in the highly

precise $O_2$ standard mixtures. The gravimetric mole fraction ($y_k$) of the constituent $k$ ($CO_2$, Ar, $O_2$, and $N_2$) was calculated using the molar mass ($M_i$) and a mole fraction ($x_{i,j}$) of the constituent $i$ ($CO_2$, Ar, $O_2$, $N_2$ and impurities) in the filled gas $j$ ($CO_2$ in Ar standard mixture, pure $O_2$, and pure $N_2$). Additionally, the mass ($m_j$) of the gases filled with the cylinder were incorporated into the Eq. (5) in accordance with ISO 6142-1:2015.



$$y_k = \frac{\sum_{j=1}^{r}\left(\frac{x_{k,j} \times m_j}{\sum_{i=1}^{q} x_{i,j} \times M_i}\right)}{\sum_{j=1}^{r}\left(\frac{m_j}{\sum_{i=1}^{q} x_{i,j} \times M_i}\right)} \qquad (5)$$

In this equation, $r$ and $q$ represent the number of source gases $j$ and constituents $i$, respectively while $x_{k,j}$ is the mole fraction of the constituent $k$ in the source gas $j$. Uncertainties ($u(y_k)$) associated with the gravimetric mole fraction

were calculated according to the law of propagation.

$$u^2(y_k) = \sum_{j=1}^{r}\sum_{i=1}^{q}\left(\frac{\partial y_k}{\partial x_{i,j}}\right)^2 \times u^2(x_{i,j}) + \sum_{i=1}^{q}\left(\frac{\partial y_k}{\partial M_i}\right)^2 \times u^2(M_i) + \sum_{j=1}^{r}\left(\frac{\partial y_k}{\partial m_j}\right)^2 \times u^2(m_j)$$

(6)

In this equation, $u(A)$ was the standard uncertainty for $A$. Gravimetric mole fractions of the constituents and their associated uncertainties in the mole fractions for the highly precise $O_2$ standard mixtures prepared in this study were calculated using Eq. (5) and Eq. (6) and they are listed in Table 2. As noted, the standard uncertainties for the constituents $N_2$, $O_2$, Ar, and $CO_2$ were 0.8 ppm to 1.0 ppm, 0.7 ppm to 0.8 ppm, 0.6 ppm to 0.7 ppm, and 0.03 ppm, respectively. Table 3 lists the contribution of each uncertainty factor to the purity of the source gases, molar masses

of the constituents, and masses of the source gases. These correspond to the square roots of the first, second, and third terms found in Eq. (6), respectively. Uncertainty factors in the gravimetric mole fractions were mainly those of the masses obtained for the source gases. Contributions from other sources of uncertainty were negligible. The purity of the source gases and molar masses of the constituents $i$, as well as the masses of the source gases and their associated standard uncertainties are described in Sections 4.1, 4.2, and 4.3.

**4.1 Purity of source gas**

Pure $O_2$, $N_2$, Ar, and $CO_2$ were used as source gases to prepare the standard $O_2$ mixtures. The mole fractions of the impurities present in the source gases and their associated standard uncertainties were determined based on the primary standard gases prepared in accordance with ISO 6142-1:2015. When the mole fraction of impurity $h$ was under detection limit ($L_h$), the mole fractions ($x_h$) and standard uncertainty ($u(x_{h,\,j})$) in the gas $j$ were calculated using the

equations $x_{h,j} = L_{h,j}/2$ and $(x_{h,j}) = L_{h,j}/2\sqrt{3}$. The calculated values for the impurities and purities of source gases are listed in Table 4.

**4.2 Molar masses of constituents**

The molar masses ($M_i$) of the source gases were calculated using the most recent atomic masses and isotopic abundances reported by the IUPAC. However, IUPAC values for the atomic masses of O and N have large standard

uncertainties because they reflect the variability present in the individual isotopic abundances of natural terrestrial matter. Using IUPAC values, the standard uncertainties for the $N_2$ and $O_2$ mole fractions were calculated to be 4 ppm. In addition, the atmospheric values of their isotopic abundances could not be used for calculating the molar masses of the source gases even though pure $O_2$ and $N_2$ were produced from air. This was because isotopically abundant O and





N in the source gases tended to deviate from the corresponding atmospheric value during the production process. Therefore, the isotopic abundances were precisely determined using mass spectrometry.

To prepare one highly precise $O_2$ standard mixture, pure $O_2$ of two 48 L cylinders were used, whereas pure $N_2$ of three or four 48 L cylinders were used. The abundances of the respective isotopes of O and N were determined based on

the ratios of $^{15}N/^{14}N$, $^{18}O/^{16}O$, and $^{17}O/^{16}O$ in each highly precise $O_2$ standard mixture. The ratios of $^{15}N/^{14}N$, $^{18}O/^{16}O$, and $^{17}O/^{16}O$ were calculated using the corresponding atmospheric values (Junk and Svec, 1958; Baertschi, 1976; Li et al., 1988; Barkan and Luz, 2005) and the ratios of the measured isotopes $\delta(^{15}N/^{14}N)$, $\delta(^{18}O/^{16}O)$, and $\delta(^{17}O/^{16}O)$ which were the deviation from the corresponding atmospheric value in each cylinder. The ratios of isotopes $\delta(^{15}N/^{14}N)$, $\delta(^{18}O/^{16}O)$, and $\delta(^{17}O/^{16}O)$ were equal to the values obtained for isotopes $\delta(^{15}N^{14}N/^{14}N^{14}N)$, $\delta(^{18}O^{16}O/^{16}O^{16}O)$, and

$\delta(^{17}O^{16}O/^{16}O^{16}O)$, since $\delta(^{18}O^{18}O/^{16}O^{16}O)$ and $\delta(^{17}O^{17}O/^{16}O^{16}O)$ tended to be much less than $\delta(^{18}O^{16}O/^{16}O^{16}O)$ and $\delta(^{17}O^{16}O/^{16}O^{16}O)$. Examples of the isotopes' abundances and their associated standards of uncertainty are shown in Table 1.

The atomic masses of N and O in the source gases, the pure $O_2$ and $N_2$ were determined with the relative standard uncertainties of 0.000029 % and 0.000006 %, respectively. It was shown that the uncertainty in the molar masses is

negligible (Table 3). Although the grade and supplier of the pure $O_2$ and $N_2$ used in this study were the same as those of the source gases used by Tohjima et al. (2005), the atomic masses (15.999366(1) for O and 14.006717 (4) for N) obtained for each element were different from Tohjima's reported values (15.999481(8) for O and 14.006677(4) for N). These differences resulted in a deviation of 0.4 ppm and 1.2 ppm for $O_2$ and $N_2$, respectively. Since this results inferred that the ratios of O and N isotopes changed due to production time, the isotopic abundances of O and N in the

source gases have to be precisely determined whenever the highly precise $O_2$ standard mixtures is prepared. On the other hand, the standard uncertainties in the atomic mass presented in an IUPAC technical report by De Laeter et al. (2003) were sufficient for further use in the case of Ar and $CO_2$ as source gases.

### 4.3 Determining the masses of the filled gases

The mass of each gas that was filled into the gravimetric cylinders was calculated using the mass differences before

and after filling. The standard uncertainty of the resultant mass was calculated by combining the standard uncertainties in the mass differences obtained for each gas before and after filling. To determine the uncertainty in the mass difference, three factors were evaluated i.e., the repeatability, $u(m_{rep})$ of the mass difference values, permeation, $u(m_{gas\ permeate})$ of the source gases during weighing, and buoyancy changes, $u(m_{buoyancy})$ due to the expansion of the gravimetric cylinder. The standard uncertainties ($u(m_{cyl})$) were defined according to


$$u^2(m_{cyl}) \ = \ u^2(m_{rep}) + u^2(m_{gas\ permeate}) + u^2(m_{buoyancy}). \qquad (7)$$

These factors are discussed in detail in Sections 4.3.1, 4.3.2, and 4.3.3.

### 4.3.1 Repeatability of the mass difference measurements

The repeatability of the weighing system was evaluated by continuously measuring the mass difference between the gravimetric and reference cylinders using the ABBA technique over three days. This is because the preparation of one highly precise $O_2$ standard mixture takes three days. The mass readings were taken after the gravimetric cylinder had



been left on the weighing system for at least a week. Using our weighing system, we also obtained density values for the surrounding ambient air for three days by carefully monitoring temperature, humidity, and pressure changes in the surrounding ambient air (Figure 4). Our findings indicated that the obtained mass difference values remained stable during the three-day experiment. The standard deviation of the mass difference values (0.82 mg) are represented as

repeatability, $u(m_{rep})$. The fact that the mass difference values were not affected by changes in the air density also indicated that buoyancy issues influencing the gravimetric cylinder were canceled out by changes simultaneously affecting the reference cylinder.

### 4.3.2 Permeation of source gases during weighing

The gravimetric and reference cylinders used in this study have diaphragm valves, which were joined to the cylinders

via pipe fittings and sealed with Teflon tape. The seal of diaphragm valves was made from PCTFE, through which gases tended to permeate quite slowly (Sturm, 2004). Since the permeation of the source gases during weighing the cylinders resulted in the evaluation error of the masses for source gases, we examined the permeability of purified air by monitoring the mass difference using the gravimetric cylinder filled with purified air at a pressure of 8 MPa. The changes in the mass difference values were measured over four months. From these results, it was determined that the

permeability was 0.013 mg day$^{-1}$. This effect was considered to be negligible because it is much lower than the repeatability. As such, the contribution of permeability ($u(m_{gas\ permeate})$) to the standard uncertainty calculations ($u(m_{cyl})$) was ignored. On the other hand, the permeation amount of the air from the cylinder over a year was calculated to be 4.7 mg. This may cause changes in the composition of the highly precise $O_2$ standard mixture if the mixture is kept for longtime, since the gas permeability depends on the gas species (Sturm, 2004).

### 4.3.3 Buoyancy effect of cylinder expansion

Oh et al. (2013) reported that the volume in the 10 L aluminum cylinders linearly increases with changes in the internal pressure, and the volume expansion was determined to be 24 ± 2 ml when the pressure difference in the cylinder was 12 MPa. Tohjima et al. (2005) reported a volume expansion of 22 ± 4 ml when the pressure difference was 10 MPa. In this study, we adopted that the volume expansion of the cylinders was 55 ± 5 ml, which was measured by a cylinder

supplier, when the pressure difference was 25 MPa. Compared to the expansion rates to pressure variations reported by Oh (2.0 ± 0.2 ml MPa$^{-1}$) (2013) and Tohjima (2.2 ± 0.4 ml MPa$^{-1}$) (2005), the rate of the cylinders was 2.2 ± 0.2 ml MPa$^{-1}$ because the factors contributing to uncertainty within these rates tended to remain constant. The pressure differences recorded before and after filling were 0.12 MPa, 2.5 MPa, and 9.4 MPa for $CO_2$ in Ar standard mixture, pure $O_2$, and pure $N_2$, respectively. These pressure differences were subsequently used to calculate buoyancy effects,

which were reported as 0.3 mg, 6.4 mg, and 23.9 mg for $CO_2$ in Ar standard mixture, pure $O_2$, and pure $N_2$, respectively. In turn, these caused changes in the gravimetric mole fraction of +0.5 ppm and −0.5 ppm for $O_2$ and $N_2$, respectively. The final mass difference values were corrected to take these changes into account. The standard uncertainties $u(m_{buoyancy})$ in linear expansion were considered to be negligible.

### 5 Validation of the Constituents in the Highly Precise $O_2$ Standard Mixtures

The $O_2$ mole fraction in the highly precise standard mixture would deviate from the gravimetric value if the mole fractions of other constituents have the deviations from the gravimetric values. The gravimetric and measured values





for the $CO_2$ mole fractions, $\delta(Ar/N_2)$, $\delta(O_2/N_2)$, and $O_2$ mole fractions were compared to validate the mole fractions of the constituents in the $O_2$ mole fractions in the highly precise $O_2$ standard mixtures. The values of $\delta(O_2/N_2)$ and $\delta(Ar/N_2)$ were the deviation from the corresponding values in the standard air on the AIST scale. Table 5 shows the measured $\delta(O_2/N_2)$ and $\delta(Ar/N_2)$ values calculated using Eq. (3) and Eq. (4), as well as the values for $\delta(^{15}N^{14}N/^{14}N^{14}N)$,

$\delta(^{17}O^{16}O/^{16}O^{16}O)$, $\delta(^{18}O^{16}O/^{16}O^{16}O)$, $\delta(^{16}O^{16}O/^{14}N^{14}N)$, $\delta(^{36}Ar/^{40}Ar)$, and $\delta(^{38}Ar/^{40}Ar)$.

**5.1 Determining the absolute ($O_2/N_2$) and ($Ar/N_2$) ratios using the AIST scale**

The absolute $O_2/N_2$ and $Ar/N_2$ ratios (($O_2/N_2$)$_{standard}$ and ($Ar/N_2$)$_{standard}$ )in the standard air on the AIST scale were calculated by substituting the gravimetric values of the $O_2/N_2$ and $Ar/N_2$ ratios (($O_2/N_2$)$_{STD}$ and ($Ar/N_2$)$_{STD}$) as listed in Table 2 into the ($O_2/N_2$)$_{sample}$ and the ($Ar/N_2$)$_{sample}$ of the Eq. (1) and Eq. (2). The values for $\delta(O_2/N_2)$ and

$\delta(Ar/N_2)$ were shown in Table 5.

The values of ($O_2/N_2$)$_{standard}$ and ($Ar/N_2$)$_{standard}$ were $0.2680869 \pm 0.0000016$ and $0.0119544 \pm 0.0000013$, respectively. On the AIST scale, these values corresponded to $\delta(O_2/N_2) = 0$ and $\delta(Ar/N_2) = 0$. Associated standard uncertainties were determined with regards to the law of propagation of uncertainty.

**5.2 $CO_2$ mole fractions and $Ar/N_2$ ratio**

Three primary standard gases were used to measure the $CO_2$ mole fractions in the highly precise $O_2$ standard mixtures. Table 2 shows the gravimetric and measured values and associated standard uncertainties. The $CO_2$ mole fractions in the cylinder labeled CPB28679, which had been prepared on 29 March 2017, were not measured. Differences between the gravimetric and measured values (obtained by subtracting the measured value from the gravimetric value) were found to range from $-0.17$ ppm to $0.03$ ppm. The gravimetric values were in line with the measured values, both of

which being within the accepted levels of uncertainty.

From these results, the mass of the $CO_2$ in Ar standard mixture was considered to be valid, since it was based on the mole fraction for the $CO_2$ utilized in this calculation. Figure 5a shows the plot of the measured $\delta(Ar/N_2)$ values versus the gravimetric $\delta(Ar/N_2)$ values, as well as the residuals of the measured $\delta(Ar/N_2)$ values that had been estimated using the line of best fit obtained using least squares method. The standard deviation of the residuals was 78 per meg. This

standard deviation represents a scatter in the gravimetric $Ar/N_2$ ratio mole fractions, since the measurement uncertainty for $\delta(Ar/N_2)$ was much smaller than the obtained standard deviation (Ishidoya and Murayama, 2014). The standard uncertainties for gravimetric $\delta(Ar/N_2)$ values ranged from 74 per meg to 77 per meg. The standard uncertainties were comparable to the standard deviation values obtained for the residuals, thus supporting that the uncertainty calculations for the constituents, Ar and $N_2$ were valid.

**5.3 $O_2$ mole fraction and $O_2/N_2$ ratio**

Figure 5b illustrates a plot of the measured $O_2$ mole fractions versus their gravimetric $O_2$ counterparts in the highly precise $O_2$ standard mixtures (Table 2), as well as the residual values, which had been determined from the fitting line obtained using least squares method. The standard deviation of the residuals shown in Figure 5b was determined to be 0.4 ppm, which was less than the standard uncertainties for the gravimetric $O_2$ mole fractions (0.7 ppm to 0.8 ppm).

Figure 5a shows a plot of the measured $\delta(O_2/N_2)$ values listed in Table 5 against the gravimetric $\delta(O_2/N_2)$ values listed in Table 2, as well as the residuals from the fitting line obtained using least squares method. The slope of the fitting line was determined to be $1.00162 \pm 0.00029$. The $\delta(O_2/N_2)$ values obtained were 0.16 % higher than those of




gravimetric $\delta(O_2/N_2)$, whereas the standard deviation of the residuals was 3.6 per meg. Since the standard uncertainties for gravimetric $\delta(O_2/N_2)$ ranged from 3.2 per meg to 4.0 per meg, the standard deviation proved to be in line with the standard uncertainties for the corresponding gravimetric values. Additionally, the results for $O_2$ mole fraction and $\delta(O_2/N_2)$ reinforced the idea that the method for calculating the uncertainties of the constituents, $O_2$ and $N_2$ was proper

and accurate. On the other hand, the measured $\delta(O_2/N_2)$ values were lower than their $\delta(^{16}O^{16}O/^{14}N^{14}N)$ counterparts by 18.2 per meg to 27.1 per meg (Table 5). The differences between the $\delta(O_2/N_2)$ and $\delta(^{16}O^{16}O/^{14}N^{14}N)$ values were larger than the standard uncertainties obtained for both values. This means that the deviation of isotopic ratios for O and N in the highly precise $O_2$ standard mixtures from the corresponding atmospheric values contributed to the $\delta(O_2/N_2)$ values obtained, even though $\delta(O_2/N_2)$ can be expressed as $\delta(^{16}O^{16}O/^{14}N^{14}N)$, especially in case of air sample

measurements.

### 6 Comparison with Previous Values

To confirm the consistency of the results obtained using the highly precise $O_2$ standard mixtures, we preliminarily compared $O_2/N_2$ ratios on both the AIST and NIES scale. Additionally, the mole fraction of atmospheric $O_2$ and Ar were determined based on the highly precise $O_2$ standard mixtures and then compared to previously reported values.

**6.1 Comparison between $O_2/N_2$ ratios on the AIST and NIES scales**

In 2015, $\delta(O_2/N_2)$ values in the air samples from Hateruma Island were measured. Twice a month, the air samples were collected in a Pyrex glass. Using these air samples, it was determined that the $\delta(O_2/N_2)$ value on the AIST scale was $-62.8 \pm 3.2$ per meg. The standard uncertainty was determined based on the standard deviation of the $\delta(O_2/N_2)$ values in air samples. Using Eq. (1), the $\delta(O_2/N_2)$ value was then converted to the absolute $O_2/N_2$ ratio by utilizing the

absolute $(O_2/N_2)_{standard}$ value on the AIST scale. In 2015, the absolute $O_2/N_2$ ratio on Hateruma Island was 0.2680761 $\pm$ 0.0000018. This absolute $O_2/N_2$ value was converted to the corresponding $\delta(O_2/N_2)$ value on the NIES scale using the Eq. (1), since the absolute $(O_2/N_2)_{standard}$ value on the NIES scale was 0.2681708 $\pm$ 0.0000036, which corresponded to the results reported by Tohjima ($\delta(O_2/N_2) = 0$) (Tohjima et al., 2005). The converted $\delta(O_2/N_2)$ value was found to be $-353$ per meg on the NIES scale.

Next, we used the equation $(\delta(O_2/N_2) = \delta\{(O_2 + Ar)/N_2\} \times (O_2 + kAr/O_2)_{ref})$ provided by Tohjima et al. (2005) to estimate the average $\delta(O_2/N_2)$ value in 2000. Here $k$ represents the sensitivity ratio Ar relative to $O_2$. They evaluated $k$ to be 1.13. From the equation, we found that the $\delta(O_2/N_2)$ value in 2000 is $-77$ per meg on the NIES scale. The $\delta\{O_2 + Ar)/N_2\}$ value was reported to be $-73$ per meg for Hateruma Island in 2000 (Tohjima et al., 2005). The $\{(O_2 + kAr)/O_2\}_{ref}$ value was also estimated to be a ratio (0.2816768/0.2681708 = 1.05036) of the $\{(O_2 + kAr)/N_2\}$ value

reported by Tohjima et al. (2005) to the absolute $(O_2/N_2)_{standard}$ value on the NIES scale. The drop in the $\delta(O_2/N_2)$ values from 2000 to 2015 was $-277 \pm 32$ per meg. In this case, the uncertainty represents a 95 % confidence interval. The average decrease in rate over this period was $19.0 \pm 2.2$ per meg yr$^{-1}$, which was slightly lower than previously reported values ($21.2 \pm 0.8$ per meg yr$^{-1}$ and $22.0 \pm 0.8$ per meg yr$^{-1}$) (Ishidoya, 2012a).

Differences between the $\delta(O_2/N_2)$ values recorded at Hateruma Island in 2000 and 2015 were compared to the

corresponding values recorded at La Jolla in 2000 and 2015. It was determined that the $\delta(O_2/N_2)$ value at La Jolla (Keeling and Manning, 2014) was $-327$ per meg. This value falls outside of the 95 % confidence interval and may



indicate the variations existing on the NIES and AIST scales. They may also imply that the slope of Scripps Institution of Oceanography (SIO) scale was higher than the actual value, since accurate verification of slope was not performed without highly precise $O_2$ standard mixtures. Additionally, other sources of error can exist. For this study, we were unable to directly compare the $O_2/N_2$ ratio or the $O_2$ mole fraction between the AIST and NIES scales. If the direct

comparison was possible, then the difference between both scales would become clear, and the slope of each scale could be verified by using the highly precise $O_2$ standard mixtures developed by our group.

**6.2 Determination of atmospheric $O_2$ and Ar mole fractions and comparison with previous data**

The mole fractions for atmospheric $O_2$ and Ar were determined based on the $\delta(O_2/N_2)$ and $\delta(Ar/N_2)$ values for air samples taken at Hateruma Island in 2015. The $\delta(O_2/N_2)$ and $\delta(Ar/N_2)$ values were −62.8 and −62.8 per meg,

respectively. Regarding the $(O_2/N_2)_{standard}$ and $(Ar/N_2)_{standard}$ ratios for the AIST scale, these values were used to calculate the $O_2/N_2$ and $Ar/N_2$ ratios using Eq. (1) and Eq. (2). In 2015, the calculated $O_2/N_2$ and $Ar/N_2$ ratios for samples from Hateruma Island were $0.2680701 \pm 0.0000013$ and $0.011953665 \pm 0.0000010$, respectively. The mole fractions of $O_2$ and Ar ($x_{O_2}$ and $x_{Ar}$) were calculated using the aforementioned $O_2/N_2$ and $Ar/N_2$ ratios by using the equations below.

$$x_{O_2} = K \times \frac{O_2/N_2}{(1+O_2/N_2+Ar/N_2)} \qquad (8)$$

$$x_{Ar} = K \times \frac{Ar/N_2}{(1+O_2/N_2+Ar/N_2)} \qquad (9)$$

In these two equations, $K$ is the sum of $N_2$, $O_2$, and Ar mole fractions in the air samples and was estimated to be

$999567.8 \pm 0.1$ ppm. To calculate this value, the mole fractions of Ne (18.18 ppm), He (5.24 ppm), $CH_4$ (1.82 ppm), Kr (1.14 ppm), $H_2$ (0.52 ppm), $N_2O$ (0.32 ppm), CO (0.15 ppm) and Xe (0.09 ppm) reported by Tohjima et al. (2005) and $CO_2$ (404.7 ppm) in 2015 were used. The $CO_2$ mole fraction was average $CO_2$ mole fraction which was measured using a mass spectrometer. The calculated $O_2$ and Ar mole fractions were $209339.1 \pm 1.1$ ppm and $9334.4 \pm 0.7$ ppm, respectively. The standard uncertainties were estimated in accordance with the law of propagation of uncertainties.

From 2000 to 2015, it was noted that the $O_2$ mole fraction in the air samples taken at Hateruma decreased by 52.9 ppm with a rate of 3.5 ppm $yr^{-1}$. In 2000, Tohjima reported an atmospheric Ar mole fraction of $9333.2 \pm 2.1$ ppm (2005), whereas the value reported for air samples collected on Korea's Anmyeon Island in 2002 and at Niwot Ridge in 2001 was $9332 \pm 3$ ppm (Park et al., 2004). Hence, our values for atmospheric Ar were in line with previously reported ones.

**7 Conclusion**

In this study, we demonstrated that the deviation of difference in mass between the gravimetric and reference cylinders is susceptible to temperature differences between these two cylinders. The contribution degree of the temperature difference was −14.3 mg $K^{-1}$. We also indicated that the variations of the mass difference values due to the temperature difference was able to be reduced to negligible levels by weighing both cylinders when the thermal equilibrium was

reached. Since the variations mainly depended on temperature differences rather than factors relating to the adsorption




phenomena (e.g., the temperature of the gravimetric cylinder and/or the humidity of the ambient air), it was thus, concluded that the changes in the mass differences were influenced solely by thermal effects.

We have developed a preparation technique for the production of highly precise $O_2$ standard mixtures with atmospheric levels of $CO_2$, Ar, $O_2$, and $N_2$. To determine the $O_2$ mole fractions with standard uncertainties of less than

1 ppm, repeatability in measuring the mass difference between the gravimetric and reference cylinders was determined. The impact of leakage or permeation of the source gases through the cylinders' valve, as well as change of buoyancy such as the expansion of the gravimetric cylinder as a factor of the cylinder's inner pressure were evaluated. Additionally, the molar masses of the $O_2$ and $N_2$ source gases were determined based on the abundance of their isotopes. The standard uncertainties gravimetrically calculated were in good agreement with the standard deviation for the

corresponding measured values. This indicates that the uncertainty calculations of the gravimetric values for the constituents performed in this study were accurate and valid.

On the basis of the highly precise $O_2$ standard mixtures, we determined the mole fractions of atmospheric Ar and $O_2$ at Hateruma Island in 2015. These values were $9334.4 \pm 0.7$ and $209339.1 \pm 1.1$ ppm, respectively. The atmospheric Ar mole fraction was in line with the values reported by Park ($9332 \pm 3$ ppm) and Tohjima ($9333.2 \pm 2.1$ ppm) (Park

et al., 2004; Tohjima et al., 2005). Our research indicated that the atmospheric $O_2$ mole fraction decreased by 52.9 ppm between 2000 and 2015 with a rate of 3.5 ppm $yr^{-1}$.

**Acknowledgments**

This study was partly supported by funding from the Global Environment Research Coordination System from the

Ministry of the Environment, Japan. We express our gratitude to Noritsugu Tsuda, Nobukazu Oda, Fujio Shimano of Global Environmental Forum, and Yasunori Tohjima of National Institute for Environmental Studies for their cooperation in collecting air samples at Hateruma Island.

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



Table 1. Isotopic composition and atomic masses of pure oxygen and nitrogen used to prepare a highly precise $O_2$ standard mixture for the cylinder labeled CPB28912.

| Isotope | Atomic mass[a,b] | Isotope abundance | | Isotope ratio of source gas[e] |
| --- | --- | --- | --- | --- |
| | | Atmosphere[a] | Source gas[a] | |
| $^{14}N$ | 14.0030740074(18) | 0.996337(4)[c] | 0.996346(4) | |
| $^{15}N$ | 15.000108973(12) | 0.003663(4)[c] | 0.003654(4) | $\delta^{15}N = -2.397 \pm 0.001$ ‰ |
| $^{16}O$ | 15.9949146223(25) | 0.9975684(9)[d] | 0.9975887(9) | |
| $^{17}O$ | 16.99913150(22) | 0.0003836(8)[d] | 0.0003818(8) | $\delta^{17}O = -4.66 \pm 0.05$ ‰ |
| $^{18}O$ | 17.9991604(9) | 0.0020481(5)[d] | 0.0020295(5) | $\delta^{18}O = -9.075 \pm 0.003$ ‰ |
| Sources | Atomic mass of nitrogen[a] | | Atomic mass of oxygen[a] | |
| Atmosphere | 14.006726(4) | | 15.999405(1) | |
| Source gases | 14.006717(4) | | 15.999366(1) | |

[a] The numbers in the parentheses represent the standard uncertainty in the last digits.

[b] The atomic mass and the standard uncertainty as determined by De Laeter et al. (2003).

[c] The abundance of the isotope and the standard uncertainty as determined using calculations for the absolute $^{15}N/^{14}N$ ratio obtained by Junk and Svec (1958).

[d] The abundance of the isotope and the standard uncertainty were calculated using $^{17}O/^{16}O = 12.08$ ‰ and $^{18}O/^{16}O = 23.88$ ‰ vs. the VSMOW as determined by Barkan and Luz (2005). The absolute isotope ratio for VSMOW and the standard uncertainty were determined by Li et al. (1988) for $^{17}O/^{16}O$ and Baertschi (1976) for $^{18}O/^{16}O$.

[e] The isotope ratio is defined as the difference in the corresponding atmospheric value (CRC00045) measured using a mass spectrometer. The numbers following the symbol ± denote the standard uncertainty.



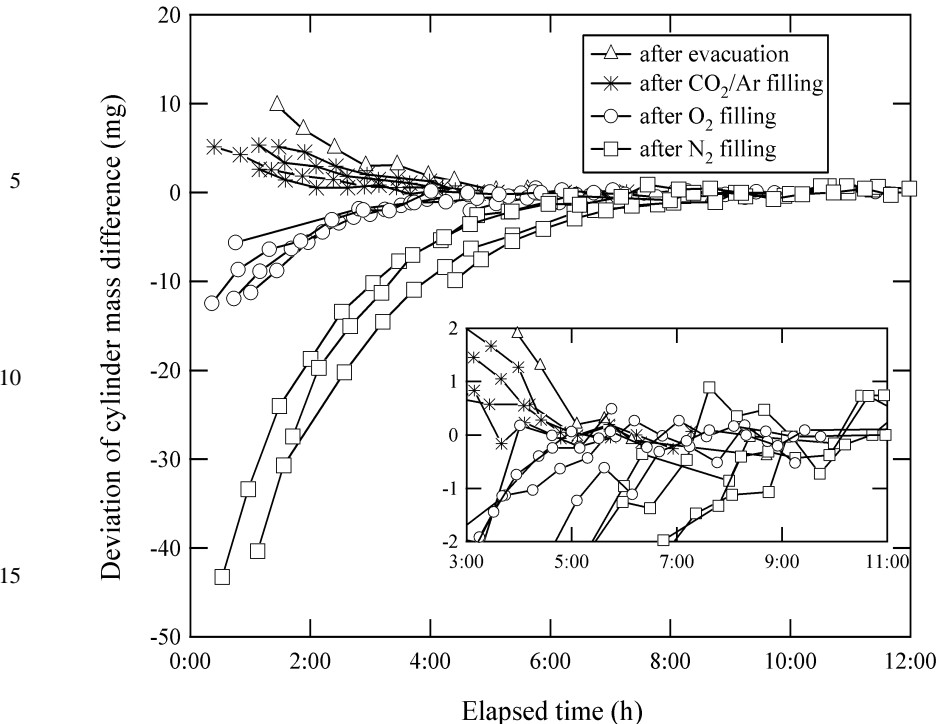

**Figure 1** Changes in the mass differences observed for the gravimetric and reference cylinders plotted against the time elapsed after evacuation of the gravimetric cylinder and filling of source gases. Masses were measured using the weighing system



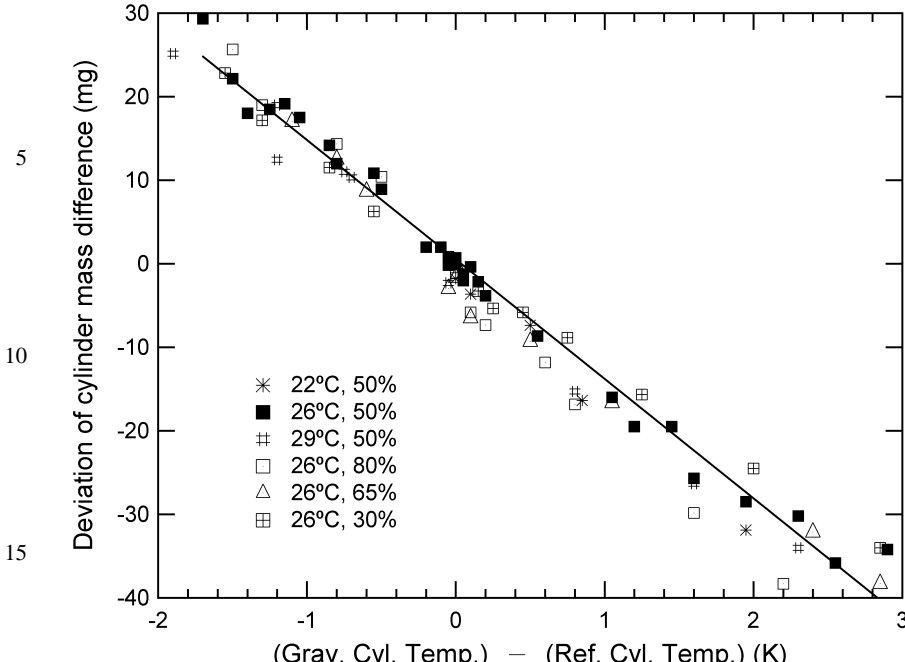

**Figure 2** Changes in the mass differences observed for the gravimetric and reference cylinders plotted against temperature differences obtained under various conditions (a temperature range from 22 ºC to 29 ºC, a humidity range from 30 % to 80 %.





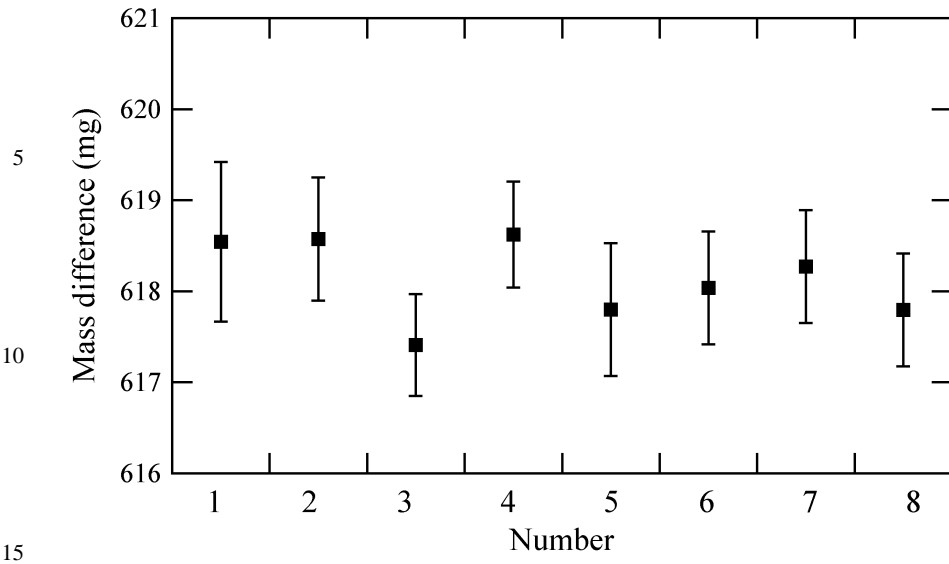

**Figure 3** Changes in the mass differences obtained for the gravimetric and reference cylinders after cylinders had been heated at 40 ℃ (Number 1 to 4) or cooled at 23 ℃ (Number 5 to 8). The error bars represent the standard uncertainty.

Table 2. Gravimetric values of $N_2$, $O_2$, and $CO_2$ mole fractions alongside $O_2/N_2$ ratios, $\delta(O_2/N_2)$, and $\delta(Ar/N_2)$, as well as the measured values obtained for $CO_2$ mole fractions from precise measurements of $O_2$ standard gases.

| Cylinder number | Preparation date | Gravimetric values[a], ppm | | | | | | Measured values, ppm |
|---|---|---|---|---|---|---|---|---|
| | | $N_2$ | $O_2$ | Ar | $CO_2$ | $(O_2/N_2)_{STD}$ | $(Ar/N_2)_{STD}$ | $\delta(O_2/N_2)$[b] | $\delta(Ar/N_2)$[b] | $CO_2$ |
| CPC00556 | 15 March 2017 | 780094.1 ± 1.0 | 210068.3 0.8 | ± 9415.2 ± 0.7 | 422.30 0.03 | ± 0.2692858 0.0000011 | ± 0.0120693 0.0000009 | ± 4471.8 ± 4.0 | 9619 ± 77 | 422.37 ± 0.14 |
| CPB28679 | 29 March 2017 | 782593.9 ± 0.8 | 207770.2 0.7 | ± 9222.1 ± 0.6 | 413.64 0.03 | ± 0.2654892 0.0000009 | ± 0.0117841 0.0000008 | ± −9689.9 ± 3.4 | −14244 ± 67 | - |
| CPB16178 | 5 April 2017 | 779014.8 ± 1.0 | 211348.4 0.8 | ± 9241.0 ± 0.7 | 395.78 0.03 | ± 0.2713021 0.0000010 | ± 0.0118624 0.0000009 | ± 11993.0 ± 4.0 | −7694 ± 77 | 395.96 ± 0.14 |
| CPB16345 | 7 April 2017 | 781499.3 ± 1.0 | 208750.7 0.8 | ± 9349.6 ± 0.7 | 400.43 0.03 | ± 0.2671156 0.0000011 | ± 0.0119636 0.0000009 | ± −3623.2 ± 4.0 | 777 ± 75 | 400.40 ± 0.14 |
| CPB16315 | 12 April 2017 | 781264.1 ± 0.9 | 209040.6 0.7 | ± 9297.0 ± 0.7 | 398.18 0.03 | ± 0.2675671 0.0000010 | ± 0.0118999 0.0000009 | ± −2595.1 ± 3.6 | −5191 ± 79 | 398.21 ± 0.14 |
| CPB16379 | 17 April 2017 | 781059.5 ± 0.8 | 209233.2 0.7 | ± 9308.6 ± 0.6 | 398.68 0.03 | ± 0.2678838 0.0000009 | ± 0.0119179 0.0000008 | ± −757.9 ± 3.3 | −3050 ± 65 | 398.68 ± 0.14 |
| CPB16349 | 13 June 2017 | 780424.7 ± 0.8 | 209813.5 0.7 | ± 9342.7 ± 0.6 | 419.06 0.03 | ± 0.2688452 0.0000009 | ± 0.0119713 0.0000008 | ± 2828.5 ± 3.4 | 1419 ± 66 | 419.22 ± 0.14 |
| CPB28912 | 15 June 2017 | 780792.3 ± 0.8 | 209437.0 0.7 | ± 9351.1 ± 0.6 | 419.44 0.03 | ± 0.2682366 0.0000009 | ± 0.0119765 0.0000008 | ± 558.1 ± 3.4 | 1851 ± 66 | 419.54 ± 0.14 |
| CPB28679 | 22 June 2017 | 780869.0 ± 0.8 | 209383.9 0.7 | ± 9328.6 ± 0.6 | 418.44 0.03 | ± 0.2681421 0.0000009 | ± 0.0119464 0.0000008 | ± 205.8 ± 3.4 | −664 ± 65 | 418.54 ± 0.14 |

[a] The numbers following the symbol ± denote the standard uncertainty.

[b] The values were calculated using absolute $O_2/N_2$ and $Ar/N_2$ in standard air as determined in Section 5.1.

Table 3. Typical contribution of each source of uncertainty (including the mass of the source gas, molar mass, and purity) to the standard uncertainties obtained for the mole fractions of $N_2$, $O_2$, Ar, and $CO_2$ in a highly precise $O_2$ standard mixture.

| Constituent | Uncertainty source, ppm | | | Combined standard uncertainty, ppm |
|---|---|---|---|---|
| | Mass of source gas | Molar mass | Purity | |
| $N_2$ | 0.77 | 0.11 | 0.05 | 0.77 |
| $O_2$ | 0.63 | 0.03 | 0.03 | 0.63 |
| Ar | 0.56 | 0.13 | 0.02 | 0.58 |
| $CO_2$ | 0.025 | 0.006 | 0.011 | 0.028 |





Table 4. Impurities in the source gases to prepare highly precise $O_2$ standard mixtures

| Impurity | Source gases, ppm | | | |
|---|---|---|---|---|
| | $CO_2$ | Ar | $O_2$ | $N_2$ |
| $N_2$ | $0.9 \pm 0.5$ | $0.12 \pm 0.07$ | $0.12 \pm 0.07$ | - |
| $O_2$ | $0.3 \pm 0.1$ | $0.5 \pm 0.3$ | - | $0.05 \pm 0.03$ |
| | | | | $0.05 \pm 0.03$ |
| Ar | - | - | $0.089 \pm 0.052$ | $0.28 \pm 0.01$ |
| | | | | $0.32 \pm 0.03$ |
| $CO_2$ | - | $0.002 \pm 0.001$ | $0.124 \pm 0.004$ | $0.002 \pm 0.001$ |
| $H_2O$ | $4.8 \pm 2.7$ | $0.05 \pm 0.03$ | $0.05 \pm 0.03$ | $0.05 \pm 0.03$ |
| $CH_4$ | $0.6 \pm 0.3$ | $0.005 \pm 0.003$ | $0.005 \pm 0.003$ | $0.005 \pm 0.003$ |
| CO | - | $0.04 \pm 0.02$ | $0.04 \pm 0.02$ | $0.04 \pm 0.02$ |
| $H_2$ | $2.2 \pm 1.3$ | - | - | - |
| Purity % | 99.99913 | 99.99993 | 99.999957 | 99.999980 |
| | | | | 99.999957 |
| | | | | 99.999954 |

The numbers following the symbol ± denote the standard uncertainty.

"-" represents the constituents which were not measured.



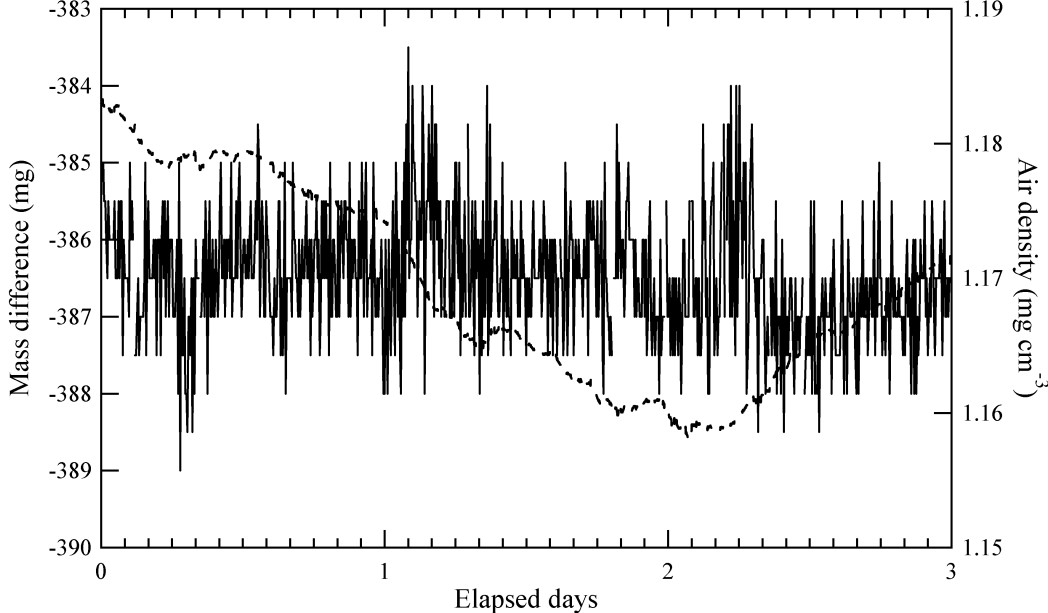

**Figure 4** Changes in the mass differences obtained for the gravimetric and reference cylinders and ambient air density for three days. The solid and dashed lines represent the mass differences and ambient air density, respectively.



Table 5. Mole fractions and standard uncertainties as determined in highly precise $O_2$ standard mixtures for $\delta({}^{15}N{}^{14}N/{}^{14}N{}^{14}N)$, $\delta({}^{17}O{}^{16}O/{}^{16}O{}^{16}O)$, $\delta({}^{18}O{}^{16}O/{}^{16}O{}^{16}O)$, $\delta({}^{16}O{}^{16}O/{}^{14}N{}^{14}N)$, $\delta(O_2/N_2)$, $\delta({}^{36}Ar/{}^{40}Ar)$, $\delta({}^{40}Ar/{}^{14}N{}^{14}N)$, and $\delta(Ar/N_2)$.

| Cylinder number | $\delta({}^{15}N{}^{14}N/{}^{14}N{}^{14}N)$ | $\delta({}^{17}O{}^{16}O/{}^{16}O{}^{16}O)$ | $\delta({}^{18}O{}^{16}O/{}^{16}O{}^{16}O)$ | $\delta({}^{16}O{}^{16}O/{}^{14}N{}^{14}N)$ | $\delta(O_2/N_2)$ | $\delta(O_2/N_2)-\delta({}^{16}O{}^{16}O/{}^{14}N{}^{14}N)$ | $\delta({}^{36}Ar/{}^{40}Ar)$ | $\delta({}^{40}Ar/{}^{14}N_2)$ | $\delta(Ar/N_2)$ |
|---|---|---|---|---|---|---|---|---|---|
| CPC00556 | −2365.0 ± 1.2 | −4032 ± 50 | −7907.8 ± 2.6 | 4477.5 ± 3.2 | 4459.2 ± 3.2 | −18.2 | −2465 ± 50 | 9649.0 ± 6.5 | 9658.1 ± 6.5 |
| CPB28679 | −2343.5 ± 1.2 | −4032 ± 50 | −8298.0 ± 2.6 | −9704.7 ± 3.2 | −9724.4 ± 3.2 | −19.7 | −1969 ± 50 | −14102.6 ± 6.5 | −14092.2 ± 6.5 |
| CPB16178 | −2372.5 ± 1.2 | −4219 ± 50 | −8279.7 ± 2.6 | 12011.7 ± 3.2 | 11991.7 ± 3.2 | −20.0 | −2197 ± 50 | −7828.0 ± 6.5 | −7818.1 ± 6.5 |
| CPB16345 | −2351.5 ± 1.2 | −4676 ± 50 | −9087.6 ± 2.6 | −3624.2 ± 3.2 | −3647.7 ± 3.2 | −23.5 | −2311 ± 50 | 712.0 ± 6.5 | 721.5 ± 6.5 |
| CPB16315 | −2356.2 ± 1.2 | −4665 ± 50 | −9069.6 ± 2.6 | −1946.8 ± 3.2 | −1970.2 ± 3.2 | −23.4 | −2228 ± 50 | −4538.2 ± 6.5 | −4528.5 ± 6.5 |
| CPB16379 | −2416.8 ± 1.2 | −4655 ± 50 | −9062.8 ± 2.6 | −763.6 ± 3.2 | −786.6 ± 3.2 | −22.9 | −2261 ± 50 | −3074.4 ± 6.5 | −3064.3 ± 6.5 |
| CPB16349 | −2407.9 ± 1.2 | −4630 ± 50 | −9036.0 ± 2.6 | 2833.1 ± 3.2 | 2810.2 ± 3.2 | −23.0 | −2360 ± 50 | 1485.7 ± 6.5 | 1495.4 ± 6.5 |
| CPB28912 | −2397.2 ± 1.2 | −4656 ± 50 | −9075.3 ± 2.6 | 554.6 ± 3.2 | 531.5 ± 3.2 | −23.2 | −2348 ± 50 | 1812.2 ± 6.5 | 1821.9 ± 6.5 |
| CPB28679 | −2390.8 ± 1.2 | −5109 ± 50 | −9941.2 ± 2.6 | 212.5 ± 3.2 | 185.4 ± 3.2 | −27.1 | −2338 ± 50 | −642.8 ± 6.5 | −633.2 ± 6.5 |

These values were calculated using the AIST scale and were given in per meg.

The numbers following the symbol ± denote the standard uncertainty.




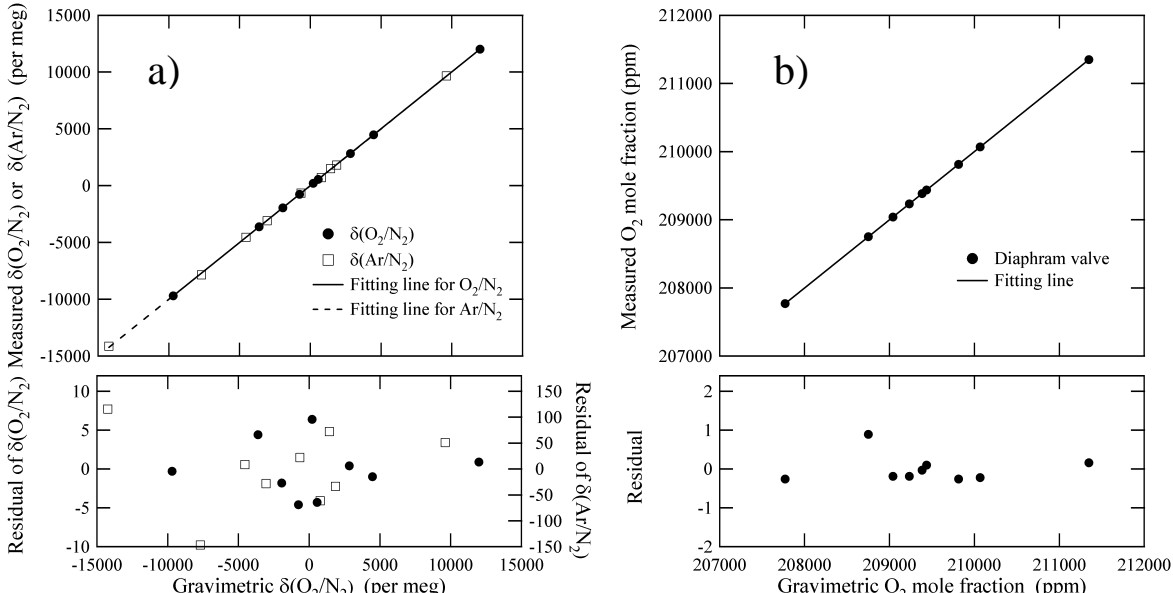

5    **Figure 5** a) The relationship between the measured and gravimetric values for $\delta(O_2/N_2)$ and $\delta(Ar/N_2)$ as determined using the AIST scale (upper). The residuals of the values for $\delta(O_2/N_2)$ and $\delta(Ar/N_2)$ from the fitting line (lower). b) The relationship between the measured and gravimetric values for $O_2$ mole fractions as measured in highly precise $O_2$ standard mixtures (upper). The residuals of the measured $O_2$ mole fraction from the fitting line (lower).

