# Peer review of "Preparation of primary standard mixtures for atmospheric oxygen measurements with less than 1 μmol mol-1 uncertainty for oxygen molar fractions"

_Atmospheric Measurement Techniques, 2018_

## Referee Comment (RC1) · Anonymous Referee #1 · 24 Jul 2018

This paper presents an improved method for preparing gravitational mixtures of O2, N2, Ar, and CO2 in air, with potential importance for a range of atmospheric measurements, particularly for detection of long-term trends in O2/N2 ratio. I sense the basic gravitational work was done with great care. But the presentation itself is not polished, and I had difficulty following some of the methods and discussions, such as the comparisons with natural air. The presentation is sufficiently unclear that it will be of limited value in documenting the method and results. There is also at least one outstanding analytical issue that may need to be addressed with further lab work. I recommend resubmission after major revision, although with the large number of substantive issues, this would be equivalent to withdrawing and resubmitting.

[Figure]

General concerns:

I can't follow the method by which the absolute mole ratios in the reference (natural) air cylinder CR00045 were assessed based on comparison the standards. This is not well explained, and seems possibly problematic. I specifically missed clarification that the mass spectrometer used to measure d(O2/N2) and d(Ar/N2) in fact measures the dominant isotopologue ratio 16O2/14N2 and Ar/14N2. Thus it should be sufficient to know the absolute 16O2/14N2 and Ar/14N2 ratios in the gravimetric standards to assess the absolute 16O2/14N2 and Ar/14N2 of CR00045 through the delta measurements. From the absolute 16O2/14N2 and Ar/14N2 ratios in CR00045, one could determine the absolute O2/N2 and Ar/N2 ratios including all isotopologues in CR00045 simply by knowing the isotopic abundances in natural air. Eqs (3) and (4), which I assume are being used in this comparison, look incorrect because they include irrelevant information on the isotopic abundances of the standard mixture. Could the authors perhaps have made the incorrect assumption that the mass spectrometer actually measures the delta based on the sum of all isotopologues?

The paper overlooks the possibility that the concentrations delivered from the tanks for analysis might differ from gravimetric ratios by either homogeneous or inhomogeneous fractionation. Numerous previous studies (e.g. Leuenberger et al., AMT 2015; Langenfelds et al, 2005, JGR -Atmospheres 110(D13); Keeling et al, JGR 1998; Keeling et al Tellus B 2004) have drawn attention to these issues, which often dominate errors and therefore cannot be ignored. As shown by both the Keeling and Leuenberger studies, a pertinent measurement is to assess the change in composition of the tank as it is depleted. This effectively is a constraint on both types of fractionation. Surface adsorption/fractionation at lower pressure ranges could be assessed by filling an evacuated tank up to modest pressure (e.g. 3 atmospheres) and looking at composition anomalies in the residual gas caused by the filling. Tests of this sort could be done with similar tanks filled with natural air, sparing the gravimetric tanks. Perhaps the authors have other ideas. In any case, some additional lab work is needed to assess

these effects, which cannot realistically be assessed theoretically.

Another omission is a discussion of the interferences from gases other than O2/N2, Ar, and CO2 on the mass spectrometer measurement. Ne, He, Kr, CH4, H2, and N2O all have abundances over 0.1 ppm in air, but presumably not in the gravimetric tanks. The effects may be small but need evaluation or discussion.

It's unclear what was learned from the paramagnetic measurements that compare gravimetric standards to a tank of synthetic air. Also, the discussion of the paramagnetic measurements lacks a discussion of interferences. I suggest that this content be cut, as it doesn't appear to address anything important.

The study lacks a direct comparison with the previous gravimetric work of Tohjima et al (2005). Section 6.1 is entitled "Comparison between O2/N2 ratios on the AIST and NIES scales", but instead of reporting such a comparison, e.g. by exchanging cylinders with NIES, this section does something else entirely: They use their measurements to report a trend in O2/N2 by combining the previous absolute estimate of O2 mole fraction at Hateruma station in Tohjima et al, with a new absolute determination at Hateruma done by the authors 15 years later. The inferred trend in O2/N2 at Hateruma is shown to be significantly smaller than the trend measured at La Jolla over the same period by the Scripps group. But before they make this very speculative comparison, they first need to carry out a direct comparison of standards. Also, I believe that the NIES group has made measurements over the full time frame at Hateruma. Surely, the NIES data should be examined before comparing with La Jolla.

Other points:

Page 2, line 21. The context of the 500 per meg figure is unclear. I assume it may reflect the decrease over some time period of measurement, but this isn't clear nor is the reason for this statement.

Page 2, lines 21-31. This paragraph is intended to provide motivation, but I found it

hard to follow. It also misses important content. I suggest this prose be replaced with a summary of current practice of calibrating O2/N2 measurements and explaining why the development of absolute standards would satisfy an important need by overcoming the reliance on the long-term stability of O2/N2 ratios in high pressure aluminum gas cylinders. Here might also be a good place to mention the relevance of homogenous and heterogeneous fractionation mechanisms and the relevance of good practice in withdrawing air from tanks. Page 3, line 30. Punctuation problem. "as such" is start of new sentence.

Page 3, line 31. Meaning of "calibration lines" is unclear to me.

Section 2.3.2. This section lacks adequate motivation. Why is it relevant to measure the O2/N2 and Ar/N2 ratios of the gravimetric mixtures when their ratios are known from the gravimetric preparation? I think the context here is a comparison with natural air. Another title for this section and few sentences of explanation are needed.

Page 5, line 11. I can't follow, as the distinction between sample and standard is unclear here. Is CRC0045 the sample or the standard? Note that the delta value for CRC00045 will be zero by definition. This is true whether the delta value is based on the dominant isotopes or not. This content therefore makes no sense to me.

Page 5, line 14 and Eq. (3) and (4). In the context of this section, it is unclear what is meant by $\delta$(O2/N2) and $\delta$(Ar/N2) without isotopic label. Does this refer to a ratio formed based on the sum of all isotopologues?

Page 5, Eq. (3) and (4). Why do 18O17O, 18O18O, 15N15N not appear in these equations?

Page 5, line 25. It would be good here to repeat that the label "standard" refers to CRC00045.

Section 2.3.3. Similar to the last section, the section title seems wrong and motivation is lacking. It's especially confusing that a comparison to synthetic air is being done.

How was the value of 20.650% determined? Since the uncertainty on 20.650% is much greater than the uncertainty on the gravimetric mixtures, it's hard to see the point of this comparison. As discussed above, I suggest cutting this section.

Page 6, lines 25 and 26. Meaning of "work" unclear. Is this meant in a thermodynamic sense? Work versus heat? Generally, this paragraph is hard to follow.

Page 6, lines 32, 33 and 36. Meaning of "equilibrium" is unclear, and is perhaps the wrong word choice. It seems it is defined operationally by the stability of the readings over time. I miss a statement about temperature measurements. How was temperature measured?

Page 7, line 16-17. "The mass difference decreases..." Unclear that this is a statement about the sign, as it reads more as a statement about magnitude, e.g. would the difference be smallest with a very large temperature difference? Would be clearer if stated as "warmer cylinders appear lighter (or heavier?)". Even on multiple readings I can't figure out which direction is implied.

Page 7, lines 21-28. The information in this paragraph should be condensed and merged with the previous paragraph. It would be easier to follow the earlier paragraph if the temperature measurements were discussed BEFORE discussing the impact on weighings.

Page 6-7, I urge that Sections 3.1 and 3.2 be merged into one section to improve readability. I note that there is no discussion of how the surface temperature of the cylinders was measured. Okay, reading further, I see it is eventually discussed. Maybe this should be mentioned above in Materials and Methods, where more detail could be given, e.g. how was thermocouple attached? Was it left in place during weighings?

Page 8, line 25. "humidity and temperature factors". If the point is that the effect is due to temperature alone, why does this sentence mention temperature factors.

Page 11. The leak-up rate of 0.013 mg/day is more than two orders of magnitude faster

than the upper bound reported in Keeling et al Tellus B, 59, 2007 for a presumably similar valve at cylinder pressure. The rate is admittedly small in the context of their application, but perhaps not in other applications, so their findings may raise concerns. They should at least cite Keeling et al and mention that the rate appears high compared to other work.

Page 8, line 28. Section 4. The header needs rewording. Suggest "Gravimetric Error Propagation"

Page 11, line 3 "Table 5 shows…" Aside from the major question I raised above about the overall logic of this calculation, I miss how the value of (16O2/14N2) standard and (40Ar/14N2) standard are assigned.

Page 12, line 37 "The d(O2/N2) values obtained were 0.16%..." I can't follow this sentence.

Page 13, lines 4-10. This looks like important information, but I can't follow. I guess this reflects my difficulty understanding the overall logic of their approach.

Page 13, line 17-18, "Using these samples…". I can't follow. The sentence appears to assume that the atmospheric O2/N2 ratio is constant. What time of year? Are these annual mean values?

Page 13, line 20. Same problem as my last comment.

Page 13, line 25. This paragraph is hard to follow. The need for Ar corrections is not explained. Wouldn't it be possible to work directly from O2/N2 measurements reported by the NIES group, who have taken care of this detail themselves? As mentioned previously, it's strange here not to directly compare gravimetric standards, so this section as a whole is problematic.

Page 14, line 1-5. As mentioned in Keeling et al (JGR, 1998), the Scripps scale factor has in fact been compared to gravimetric standards.

Section 6.2. I can't follow why this information is being presented and how it differs from material in the previous section. For example, between this and the previous section, two inconsistent values (0.2680761 and 0.2680701) for the $O_2/N_2$ ratio at Hateruma are reported for 2015. Confusing.

Page 14, line 25. "From 2000 to 2015, it was noted. . ." The basis for this estimate is not clear. Also, to report $O_2$ changes in ppm risks causing confusion unless some context is given. Does this mole fraction basis include $CO_2$? How does this estimate compare to one based on combining information on the change in $O_2/N_2$ with known changes in $CO_2$ abundance?

Page 15, line 15. See early comment about this reported rate. Needs context to avoid misunderstanding.

Figure 1. It's unclear why these curves converge to zero. If the data shown is the change relative to the last point, this should be explained in the caption.

Figure 4. Unclear which curve goes with which axis.

Figure 5a. The x axis is labeled $O_2/N_2$, but could it actually be showing both $O_2/N_2$ and $Ar/N_2$?

Table 2. This table is garbled. Some cells and some column headers appear to have inappropriate line breaks. The rows don't line up properly and the +/- symbols are often not located properly.

---

## Referee Comment (RC2) · Anonymous Referee #2 · 6 Aug 2018

The comment was uploaded in the form of a supplement:
https://www.atmos-meas-tech-discuss.net/amt-2018-192/amt-2018-192-RC2-supplement.pdf

---

## Author Response (AR1)

We wish to express our appreciation for your significant and useful comments. We have revised the manuscript, considering your comments and suggestions.

Referee #1

This paper presents an improved method for preparing gravitational mixtures of O2, N2, Ar, and CO2 in air, with potential importance for a range of atmospheric measurements, particularly for detection of long-term trends in O2/N2 ratio. I sense the basic gravitational work was done with great care. But the presentation itself is not polished, and I had difficulty following some of the methods and discussions, such as the comparisons with natural air. The presentation is sufficiently unclear that it will be of limited value in documenting the method and results. There is also at least one outstanding analytical issue that may need to be addressed with further lab work. I recommend resubmission after major revision, although with the large number of substantive issues, this would be equivalent to withdrawing and resubmitting.

General concerns:

I can't follow the method by which the absolute mole ratios in the reference (natural) air cylinder CR00045 were assessed based on comparison the standards. This is not well explained, and seems possibly problematic. I specifically missed clarification that the mass spectrometer used to measured (O2/N2) and d(Ar/N2) in fact measures the dominant isotopologue ratio 16O2/14N2 and Ar/14N2. Thus it should be sufficient to know the absolute 16O2/14N2 and Ar/14N2 ratios in the gravimetric standards to assess the absolute 16O2/14N2 and Ar/14N2 of CR00045 through the delta measurements. From the absolute 16O2/14N2 and Ar/14N2 ratios in CR00045, one could determine the absolute O2/N2 and Ar/N2 ratios including all isotopologues in CR00045 simply by knowing the isotopic abundances in natural air. Eqs (3) and (4), which I assume are being used in this comparison, look incorrect because they include irrelevant information on the isotopic abundances of the standard mixture. Could the authors perhaps have made the incorrect assumption that the mass spectrometer actually measures the delta based on the sum of all isotopologues?

**Response:** The absolute values which were precisely determined by the gravimetric method were the (O2/N2) and (Ar/N2) ratios not the (16O2/14N2) and (40Ar/14N2) ratios. Additionally, if the absolute (O2/N2) is calculated based on the (16O2/14N2), its uncertainty is larger than the gravimetrically calculated uncertainty. Therefore, we discussed the d(O2/N2) and d(Ar/N2) ratios based on the absolute (O2/N2) and (Ar/N2) ratios not isotopologue ratios 16O2/14N2 and 40Ar/14N2. We revised the sentence to be easy to be understood.(section 2.4.1, 5.1)

The paper overlooks the possibility that the concentrations delivered from the tanks for analysis might differ from gravimetric ratios by either homogeneous or inhomogeneous fractionation. Numerous previous studies (e.g. Leuenberger et al., AMT 2015; Langenfelds et al, 2005, JGR -Atmospheres 110(D13); Keeling et al, JGR 1998; Keeling et al Tellus B 2004) have drawn attention to these issues, which often dominate errors and therefore cannot be ignored. As shown by both the Keeling and Leuenberger studies, a pertinent measurement is to assess the change in composition of the tank as it is depleted. This effectively is a constraint on both types of fractionation. Surface adsorption/fractionation at lower pressure ranges could be assessed by filling an evacuated tank up to modest pressure (e.g. 3 atmospheres) and looking at composition anomalies in the residual gas caused by the filling. Tests of this sort could be done with similar tanks filled with natural air, sparing the gravimetric tanks. Perhaps the authors have other ideas. In any case, some additional lab work is needed to assess these effects, which cannot realistically be assessed theoretically. Another omission is a discussion of the interferences from gases other than O2/N2, Ar, and CO2 on the mass spectrometer measurement. Ne, He, Kr, CH4, H2, and N2O all have abundances over 0.1 ppm in air, but presumably not in the gravimetric tanks. The effects may be small but need evaluation or discussion.

**Response:** we used the same type of the cylinders which Tohjima et al. had used. Since they had already verified the change of the concentrations delivered from the tanks for analysis, we didn't perform this verification. However, because we didn't discuss the verification in this paper, we add the sentences about their verification (P13, L14−L18). We carried out an additional experiment for the interferences from Ne and added the result in the paper (P12, L17−L22), since the molar fraction of Ne is highest in the minor components.

It's unclear what was learned from the paramagnetic measurements that compare gravimetric standards to a tank of synthetic air. Also, the discussion of the paramagnetic measurements lacks a discussion of interferences. I suggest that this content be cut, as it doesn't appear to address anything important.

**Response:** We removed this content according to your comment

The study lacks a direct comparison with the previous gravimetric work of Tohjima et al (2005). Section 6.1 is entitled "Comparison between O2/N2 ratios on the AIST and NIES

scales", but in stead of reporting such a comparison,e.g. by exchanging cylinders with NIES, this section does something else entirely: They use their measurements to report a trend in O2/N2 by combining the previous absolute estimate of O2 mole fraction at Hateruma station in Tohjima et al, with a new absolute determination at Hateruma done by the authors 15 years later. The inferred trend in O2/N2 at Hateruma is shown to be significantly smaller than the trend measured at La Jolla over the same period by the Scripps group. But before they make this very speculative comparison, they first need to carry out a direct comparison of standards. Also, I believe that the NIES group has made measurements over the full time frame at Hateruma. Surely, the NIES data should be examined before comparing with La Jolla. Other points: Page 2, line 21. The context of the 500 per meg figure is unclear. I assume it may reflect the decrease over some time period of measurement, but this isn't clear nor is the reason for this statement.

**Response:** We revised to the comparison between the O2/N2 ratios at Hateruma in 2015 determined by AIST and by NIES. Now, a direct comparison between NIES scale and AIST scale using gravimetric standard gases is being performed. In other paper, we will present detail of the results (section 6.1).

Page 2, lines 21-31. This paragraph is intended to provide motivation, but I found it hard to follow. It also misses important content. I suggest this prose be replaced with a summary of current practice of calibrating O2/N2 measurements and explaining why the development of absolute standards would satisfy an important need by overcoming the reliance on the long-term stability of O2/N2 ratios in high pressure aluminum gas cylinders. Here might also be a good place to mention the relevance of homogenous and heterogeneous fractionation mechanisms and the relevance of good practice in withdrawing air from tanks.

**Response:** We revised to explaining why the development of absolute standards would satisfy an important need by overcoming the reliance on the long-term stability of O2/N2 ratios in high pressure aluminum gas cylinders in accordance with your comments (P2, L20–P3, L9)

Page 3, line 30. Punctuation problem. "as such" is start of new sentence.
**Response:** We revised the sentence (P4, L25).
Page 3, line 31. Meaning of "calibration lines" is unclear to me.
**Response:** We revised the sentence from "calibration lines" to "the relation between the outputs of mass comparators and the masses of artifacts".

Section 2.3.2. This section lacks adequate motivation. Why is it relevant to measure the O2/N2 and Ar/N2 ratios of the gravimetric mixtures when their ratios are known from the gravimetric preparation? I think the context here is a comparison with natural air. Another title for this section and few sentences of explanation are needed.

**Response:** We add the motivation in section 2.4.1 (p5, L26-L27)

Page 5, line 11. I can't follow, as the distinction between sample and standard is unclear here. Is CRC0045 the sample or the standard? Note that the delta value for CRC00045 will be zero by definition. This is true whether the delta value is based on the dominant isotopes or not. This content therefore makes no sense to me.

**Response:** CRC0045 is used as the reference air not sample air. The section 2.3 were revised overall (moving from the section 2.3 to the section 2.4).

Page 5, line 14 and Eq. (3) and (4). In the context of this section, it is unclear what is meant by δ(O2/N2)and δ(Ar/N2) without isotopic label. Does this refer to a ratioformed based on the sum of all isotopologues?

Page 5, Eq. (3) and (4). Why do 18O17O, 18O18O, 15N15N not appear in these equations?

Page 5, line 25. It would be good here to repeat that the label "standard" refers to CRC00045.

**Response:** The section 2.3 were revised overall (moving from the section 2.3 to the section 2.4).

Section 2.3.3. Similar to the last section, the section title seems wrong and motivation is lacking. It's especially confusing that a comparison to synthetic air is being done.
How was the value of 20.650% determined? Since the uncertainty on 20.650% is much greater than the uncertainty on the gravimetric mixtures, it's hard to see the point of this comparison. As discussed above, I suggest cutting this section.

**Response:** We removed this section.

Page 6, lines 25 and 26. Meaning of "work" unclear. Is this meant in a thermodynamic sense? Work versus heat? Generally, this paragraph is hard to follow.

**Response:** We revised the sentences (P7, L14 –L23).

Page 6, lines 32, 33 and 36. Meaning of "equilibrium" is unclear, and is perhaps the wrong word choice. It seems it is defined operationally by the stability of the readings overtime. I miss a statement about temperature measurements. How was temperature measured?

**Response:** The "equilibrium" mean thermal and water adsorption equilibrium for the surface of the sample cylinder (P7, L25). We add the statement about temperature measurements according to your comments (P4, L34 –L36)

Page7,line16-17. "The mass difference decreases..."Unclear that this is a statement about the sign, as it reads more as a statement about magnitude, e.g. would the difference be smallest with a very large temperature difference? Would be clearer if stated as "warmer cylinders appear lighter (or heavier?)". Even on multiple readings I can't figure out which direction is implied.

**Response:** We add the sentence according to your comments (P8, L2 –L3)

Page 7, lines 21-28. The information in this paragraph should be condensed and merged with the previous paragraph. It would be easier to follow the earlier paragraph if the temperature measurements were discussed BEFORE discussing the impact on weighings.

**Response:** We merged this paragraph and the previous paragraph and discuss the temperature measurements before discussing the impact on weighing ( P7, L31–P8, L3)

Page 6-7, I urge that Sections 3.1 and 3.2 be merged into one section to improve readability. I note that there is no discussion of how the surface temperature of the cylinders was measured. Okay, reading further, I see it is eventually discussed. Maybe this should be mentioned above in Materials and Methods, where more detail could be given, e.g. how was thermocouple attached? Was it left in place during weighings?

**Response:** Section 3 was revised overall. The method to measure the cylinder's temperature was mentioned in Materials and Methods.

Page 8, line 25. "humidity and temperature factors". If the point is that the effect is due to temperature alone, why does this sentence mention temperature factors.

**Response:** We mistakes the sentence. Thermal effect is due to temperature difference alone. The sentence were revised (P9, L2 –L3)
.

Page11. The leak-uprate of 0.013mg/day is more than two orders of magnitude faster than the upper bound reported in Keeling et al Tellus B, 59, 2007 for a presumably similar valve at cylinder pressure. The rate is admittedly small in the context of their application, but perhaps not in other applications, so their findings may raise concerns. They should at least cite Keeling et al and mention that the rate appears high compared to other work.

**Response:** The leak rate we measured was calculated from monitoring mass of leakage gas. The value reported by Keeling et al. is the change rate of O2/N2 ratio. Both value cannot be compared.

Page 8, line 28. Section 4. The header needs rewording. Suggest "Gravimetric Error Propagation"

**Response:** The header replaced from "preparation of the O2 standard mixtures" to "Evaluation of uncertainty factors for the O2 standard mixtures".

Page 12, line 3 "Table 5 shows..." Aside from the major question I raised above about the overall logic of this calculation, I miss how the value of (16O2/14N2) standard and (40Ar/14N2) standard are assigned.

**Response:** We explained above about this.

Page 12, line 37 "The d(O2/N2) values obtained were 0.16%..." I can't follow this sentence.

**Response:** We revised the sentence to easily understand it (P13, L5-L7).

Page 13, lines 4-10. This looks like important information, but I can't follow. I guess this reflects my difficulty understanding the overall logic of their approach.

Page 13, line 17-18, "Using these samples...". I can't follow. The sentence appears to assume that the atmospheric O2/N2 ratio is constant. What time of year? Are these annual mean values?

**Response:** We revised the sentence overall to easily understand the overall logic (P13, L7–L13).

Page 13, line 25. This paragraph is hard to follow. The need for Ar corrections is not explained. Wouldn't it be possible to work directly from O2/N2 measurements reported by the NIES group, who have taken care of this detail themselves? As mentioned previously, it's strange here not to directly compare gravimetric standards, so this section as a whole is problematic.

**Response:** The section 6.1 was revised overall. Our value was compared with annual average in 2015 reported by the NIES group.

Page 14, line 1-5. As mentioned in Keeling et al (JGR, 1998), the Scripps scale factor has in fact been compared to gravimetric standards.
**Response:** We removed the Scripps data.

Section6.2. I can't follow why this information is being presented and how it differs from material in the previous section. For example, between this and the previous section, two inconsistent values (0.2680761 and 0.2680701) for the O2/N2 ratio at Hateruma are reported for 2015. Confusing.
**Response:** We cannot completely verify the absolute values in the highly precise O2 standard mixtures (HPO), because there is no standard mixture with uncertainty to be able to verify the HPOs. A method unlike the method performed in the section 5 is considered to be necessary. Additionally, we think that the validation of absolute values is scientifically important to enable the comparison with a previous study, for example, O2 molar fraction (0.20946) determined by Machta and Hughes(1970), etc. We revised in consistent values according to your comments.

Page14, line25. "From 2000 to 2015, it was noted..."The basis for this estimate is not clear. Also, to report O2 changes in ppm risks causing confusion unless some context is given. Does this mole fraction basis include CO2? How does this estimate compare to one based on combining information on the change in O2/N2 with known changes in CO2 abundance?
**Response:** We removed the sentence.

Page 15, line 15. See early comment about this reported rate. Needs context to avoid misunderstanding.
**Response:** We removed the sentence.

Figure 1. It's unclear why these curves converge to zero. If the data shown is the change relative to the last point, this should be explained in the caption.
**Response:** We explained the point in the caption.

Figure 4. Unclear which curve goes with which axis.

**Response:** We revised Figure 4.

Figure 5a. The x axis is labeled O2/N2, but could it actually be showing both O2/N2 and Ar/N2?

**Response:** We revised Figure 5.

Table 2. This table is garbled. Some cells and some column headers appear to have inappropriate line breaks. The rows don't line upproperly and the +/- symbols are often not located properly.

**Response:** We revised Table 2.

Reviewer #2:

This paper describes an improved method for preparing synthetic gas mixtures of oxygen in artificial air by gravimetry (weighing). The use of a new mass comparator in the automatic weighing system and a thorough uncertainty evaluation allows for a suite of mixtures that have exceptionally low uncertainties. These have been verified with high level analytical methods that show a very good consistency within the suite and with other/previous high level standard mixtures. Nevertheless there are some principal comments and specific issues that need to be revised. I therefore recommend resubmission after major revisions.

General comments

The metrics and terminology lack to some extent concordance with international recommendations, standards and good practice. Even if some quantity and unit 'habits' are well established in atmospheric science, they are not to be taken as a role model because they are very often source of misunderstanding and misconception. Some xamples are given in the following points:

1. The use of 'mole fraction' as a quantity denomination is depreciated and should be replaced by 'amount (of substance) fraction' or 'molar fraction'. Derived quantities should be defined by quantities and not by units (mole is a unit). Angles can be defined as 'length ratios' and not as 'meter ratios'. A mass fraction is not called gram fraction either. 'Mixing ratio' or 'atomic weight' are established use of quantity denominations but misleading because they mean 'molar fraction' and 'atomic mass'. Further literature is ISO 80000-9, IUPAC gold book, T. Cvitas, metrologia 2003.

**Response:** We revised from mole fraction to molar fraction in accordance with your comments

2. The use of the unit ppm for $\mu mol/mol$ is also depreciated because it is not obvious if it is a relative or absolute unit. Please keep $\mu mol/mol$, it is not that long.

**Response:** We kept $\mu mol/mol$ in this paper in accordance with your comments

3. The definition of $\delta$ ($O_2/N_2$) in 'per meg' is misleading because it contains the factor $10^6$ (equations 1 to 4). All indications in 'per meg' are redundant but need a mention of the standard. We would prefer to omit this notation or use it correctly. See also Coplen (DOI: 10.1002/rcm.5129) Note 7 page 2541 and Milton et al. (DOI: 10.1002/rcm.836)

**Response:**. We revised the equation 1 to 4 in accordance with your comments

.

The aspects of pressure dependent adsorption and desorption of analytes inside the pressurised cylinders is not discussed but may be relevant for interpreting results of certain gases (carbon dioxide).

**Response:**. We added the sentences for aspects of pressure dependent adsorption and desorption in this paper (P13, L14L18).

The issue of analytical interference when comparing standards to real air samples is not discussed but may also be relevant (water-issue).

**Response:**. We added the sentences of the interferences in this paper (P12, L17-22)

.

Specific comments:

Page 1, line 3: Replace mole fraction by molar fraction (throughout the text)

**Response:**. We replace mole fraction by molar fraction.

Page 1, line 4: Correct name Matsumoto

**Response:**. We revised the name.

Page 1, line 10: Omit per meg information in the abstract without introduction and replace ppm by μmol/mol

**Response:**. We omited per meg in the abstract without introduction and replaced ppm by μmol/mol according to your comment.

Page 2 line 2: omit (per meg) and '$\times 10^6$' and in equations 2 to 4)

**Response:**. We revised equation 2 to 4 according to your comment.

Page 2 line 24: use linear calibration function instead of calibration line (all instances)

Page 2 line 31: word order: … have not yet been …

**Response:**. This sentence was removed.

Page 2 line 33: Replace weight measurement by mass measurement (you indicate mg which is the unit of mass and not N which would be the unit of weight (gravitational force))

**Response:**. We replace weight measurement by mass measurement in accordance with your comment (P3, L13

Page 3 line 2: … were validated …

**Response:**. We revised the word according to your comment.(P3, L17)

Page 3 line 26: the expression of 'gravimetric cylinder' is misleading (further instances). In fact it is the cylinder containing the gravimetrically prepared mixture. Be clear in describing the procedure.

**Response:**. We revised the sentences of 'gravimetric cylinder' to 'sample cylinder' through this paper.

Page 3 line 36:   … were traced to the International …

**Response:**. We revised the sentence according to your comment (P4 L31)

Page 4 line 14: these may not be ratios of CO2 to Ar but molar fractions?

**Response:**. We replaced ratios of CO2 to Ar by molar ratios of CO2 to Ar (P4,L1-L2)

Page 6 line 8: … factors of uncertainty…

**Response:**. We revised the sentence according to your comment (P6, L32).

Page 6 line 24: Sentence difficult to understand. Please rephrase

**Response:**. We revised the sentences according to your comment (P7, L14-L16)

Page 12 line 6: Why are the ratios absolute? Is there a convention to reference to AIST

**Response:**. We revised the caption of section 5.1.

Table 1 last column: the isotope ratios should be expressed as …= (x.xxx ± y.yyy) ‰

**Response:**. We expressed the isotope ratios as …= (x.xxx ± y.yyy) ‰

Table 2 is hardly readable. Please rearrange for better reading.

**Response:**. We rearranged the table 2

Table 5 title: The indicated numbers represent ratios not fractions
**Response:**. We revised from fractions to ratios

Figure 5 a: The x-axis concerns also Ar/N2.
**Response:**. We revised the x-axis according to your comment

[revised manuscript text omitted]
(O_2/N_2)_{HPO\_grav}$ ratio and the $\delta(Ar/N_2)_{HPO\_grav}$ ratio. The values of $\delta(O_2/N_2)_{HPO\_meas}$ and $\delta(Ar/N_2)_{HPO\_meas}$ were calculated using mass-spectrometry based on isotopic ratios $^{15}N^{14}N/^{14}N^{14}N$, $^{17}O^{16}O/^{16}O^{16}O$, $^{18}O^{16}O/^{16}O^{16}O$, $^{36}Ar/^{40}Ar$, and $^{38}Ar/^{40}Ar$ as depicted in equations (3) and (4). Isotopic species of $^{17}O^{17}O$, $^{18}O^{17}O$, $^{18}O^{18}O$, $^{15}N^{15}N$, were negligible because the abundance of these species was very small.

$$\delta(\mathrm{O_2/N_2})_{\mathrm{HPO\_meas}} = \left[\delta(^{16}\mathrm{O}^{16}\mathrm{O}/\,^{14}\mathrm{N}\,^{14}\mathrm{N})_{\mathrm{HPO\_meas}} + 1\right] \times$$

$$\left[\frac{1+\,^{17}\mathrm{O}^{16}\mathrm{O}/^{16}\mathrm{O}^{16}\mathrm{O}+^{18}\mathrm{O}^{16}\mathrm{O}/^{16}\mathrm{O}^{16}\mathrm{O}}{1+\,^{15}\mathrm{N}\,^{14}\mathrm{N}/^{14}\mathrm{N}^{14}\mathrm{N}}\right]_{\mathrm{HPO}} \Bigg/ \left[\frac{1+\,^{17}\mathrm{O}^{16}\mathrm{O}/^{16}\mathrm{O}^{16}\mathrm{O}+^{18}\mathrm{O}^{16}\mathrm{O}/^{16}\mathrm{O}^{16}\mathrm{O}}{1+\,^{15}\mathrm{N}\,^{14}\mathrm{N}/^{14}\mathrm{N}^{14}\mathrm{N}}\right]_{\mathrm{ref}} - 1 \quad (3)$$

$$\delta(\mathrm{Ar/N_2})_{\mathrm{HPO\_meas}} = \left[\delta(^{40}\mathrm{Ar}/\,^{14}\mathrm{N}^{14}\mathrm{N})_{\mathrm{HPO\_meas}} + 1\right] \times \left[\frac{1+\,^{36}\mathrm{Ar}/^{40}\mathrm{Ar}+^{38}\mathrm{Ar}/^{40}\mathrm{Ar}}{1+\,^{15}\mathrm{N}\,^{14}\mathrm{N}/^{14}\mathrm{N}^{14}\mathrm{N}}\right]_{\mathrm{HPO}} \Bigg/ \left[\frac{1+\,^{36}\mathrm{Ar}/^{40}\mathrm{Ar}+^{38}\mathrm{Ar}/^{40}\mathrm{Ar}}{1+\,^{15}\mathrm{N}\,^{14}\mathrm{N}/^{14}\mathrm{N}^{14}\mathrm{N}}\right]_{\mathrm{ref}} - 1$$

$$(4)$$

The values of $^{15}\mathrm{N}^{14}\mathrm{N}/^{14}\mathrm{N}^{14}\mathrm{N}$, $^{17}\mathrm{O}^{16}\mathrm{O}/^{16}\mathrm{O}^{16}\mathrm{O}$, and $^{18}\mathrm{O}^{16}\mathrm{O}/^{16}\mathrm{O}^{16}\mathrm{O}$ in the HPOs and the AIST reference air were calculated using isotope abundances of O and N determined by the procedure described in section 2.3 (Table 1). The $^{36}\mathrm{Ar}/^{40}\mathrm{Ar}$ ratio of pure Ar filled in the HPOs was calculated using equation $^{36}\mathrm{Ar}/^{40}\mathrm{Ar} = [\delta(^{36}\mathrm{Ar}/^{40}\mathrm{Ar})_{\mathrm{HPO\_meas}} + 1] \times (^{36}\mathrm{Ar}/^{40}\mathrm{Ar})_{\mathrm{ref}}$. The $\delta(^{36}\mathrm{Ar}/^{40}\mathrm{Ar})_{\mathrm{HPO\_meas}}$ value was determined by mass spectrometry of the HPOs. The $(^{36}\mathrm{Ar}/^{40}\mathrm{Ar})_{\mathrm{ref}}$ value obtained was the atmospheric value ($^{36}\mathrm{Ar}/^{40}\mathrm{Ar} = 0.003349 \pm 0.000004$), because isotopic abundances of Ar in the AIST reference air were equal to that of the atmospheric value. The value of $^{38}\mathrm{Ar}/^{40}\mathrm{Ar}$ in the HPOs and the AIST reference air, which could not be measured, was assumed to be $^{38}\mathrm{Ar}/^{40}\mathrm{Ar} = 0.000631 \pm 0.000004$ taken from previous reports as the atmospheric values. Deviations of respective abundances of $^{38}\mathrm{Ar}$ from the atmospheric value were considered to be less than the uncertainty of the atmospheric value for $^{38}\mathrm{Ar}$. The atmospheric values of isotopic abundances for Ar were reported in an IUPAC technical report (Böhlk, 2014).

On the other hand, the absolute $\mathrm{O_2/N_2}$ ratio in the AIST reference air was calculated by substituting the $(\mathrm{O_2/N_2})_{\mathrm{HPO\_grav}}$ in the HPOs and the $\delta(\mathrm{O_2/N_2})_{\mathrm{HPO\_meas}}$ for $(\mathrm{O_2/N_2})_{\mathrm{sam}}$ and for $\delta(\mathrm{O_2/N_2})$ in equation (1). The absolute $\mathrm{Ar/N_2}$ ratio in the AIST reference air was calculated in same manner (see the section 5.3).

**2.4.2 Measurements of $CO_2$ in highly precise $O_2$ standard mixtures**

Molar fractions of $CO_2$ in HPOs were verified using a cavity ring-down spectrometer (G2301, Picarro Inc., USA) equipped with a multi-port valve (Valco Instruments Co. Inc., USA) for gas introduction and a mass flow controller (SEC-N112, 100SCCM, Horiba STEC, CO., Ltd, Japan). Molar fractions were determined using three primary standard gases (364.50 ± 0.14 µmol mol$^{-1}$, 494.04 ± 0.14 µmol mol$^{-1}$, and 500.32 ± 0.14 µmol mol$^{-1}$) that had been prepared from pure $CO_2$ and purified Air (G1 grade, Japan Fine Products, Japan) in accordance with ISO 6142-1:2015. The  pure $CO_2$  was  the same as the source gas used for preparation of the HPOs.
* * *
$$\delta(\mathrm{Ar/N_2})\,(\text{per meg}) = \left[\frac{(\mathrm{Ar/N_2})_{\text{sample}}}{(\mathrm{Ar/N_2})_{\text{standard}}} - 1\right] \times 10^6 \qquad (2)$$

where the subscripts "sample" and "standard" refer to the sample air and standard air in the same way as $\delta(O_2/N_2)$, respectively. In this study, natural air in 48 L aluminum cylinder (Cylinder No. CRC00045), equipped with a diaphragm valve (G-55, Hamai Industries Limited, Japan) was used as the standard air to determine the $\delta(O_2/N_2)$ and $\delta(Ar/N_2)$ values on the AIST scale (Ishidoya and Murayama, 2014). The mass spectrometer was adapted to simultaneously measure ion beam currents for masses 28 ($^{14}N^{14}N$), 29 ($^{15}N^{14}N$), 32 ($^{16}O^{16}O$), 33 ($^{17}O^{16}O$), 34 ($^{18}O^{16}O$), 36 ($^{36}Ar$), 40 ($^{40}Ar$), and 44 ($^{12}C^{16}O^{16}O$). These masses were also noted as deviations in $\delta(^{15}N^{14}N/^{14}N^{14}N)$, $\delta(^{17}O^{16}O/^{16}O^{16}O)$, $\delta(^{18}O^{16}O/^{16}O^{16}O)$, $\delta(^{16}O^{16}O/^{14}N^{14}N)$, $\delta(^{36}Ar/^{40}Ar)$, $\delta(^{40}Ar/^{14}N_2)$, and $\delta(^{12}C^{16}O^{16}O/^{14}N^{14}N)$ from the corresponding atmospheric values that had been recorded for the standard air.

In the case of sample air, it was assumed that both the $\delta(O_2/N_2)$ and $\delta(Ar/N_2)$ values were equal to those of $\delta(^{16}O^{16}O/^{14}N^{14}N)$ and $\delta(^{40}Ar/^{14}N^{14}N)$, since the ratios of Ar, O, and N isotopes present in the atmosphere tended to be spatiotemporally constant. On the other hand, the isotopic ratios of pure Ar, $O_2$, and $N_2$ used in this study were different from the atmospheric values listed in Table 1. Consequently, both the $\delta(O_2/N_2)$ and $\delta(Ar/N_2)$ values in the highly precise $O_2$ standard mixtures were computed using the measurements obtained for $^{15}N^{14}N/^{14}N^{14}N$, $^{17}O^{16}O/^{16}O^{16}O$, $^{18}O^{16}O/^{16}O^{16}O$, $^{36}Ar/^{40}Ar$, and $^{38}Ar/^{40}Ar$, as depicted in the equations below.

$$\delta(O_2/N_2) = \left\{ \frac{\left(\frac{^{16}O^{16}O/\,^{14}N_2}{}\right)_{STD}}{\left(\frac{^{16}O_2/\,^{14}N_2}{}\right)_{standard}} \times \left[\frac{1+\,^{17}O^{16}O/^{16}O^{16}O+\,^{18}O^{16}O/^{16}O^{16}O}{1+\,^{15}N^{14}N/^{14}N^{14}N}\right]_{STD} \middle/ \left[\frac{1+\,^{17}O^{16}O/^{16}O^{16}O+\,^{18}O^{16}O/^{16}O^{16}O}{1+\,^{15}N^{14}N/^{14}N^{14}N}\right]_{standard} - 1 \right\} \times$$

$$10^6 \hspace{10cm} (3)$$

$$\delta(Ar/N_2) = \left\{ \frac{\left(\frac{^{40}Ar/\,^{14}N^{14}N}{}\right)_{STD}}{\left(\frac{^{40}Ar/\,^{14}N^{14}N}{}\right)_{
[revised manuscript text omitted]
 5.  δ($^{15}$N$^{14}$N/$^{14}$N$^{14}$N)$_{HPO\_meas}$, δ($^{17}$O$^{16}$O/$^{16}$O$^{16}$O)$_{HPO\_meas}$, δ($^{18}$O$^{16}$O/$^{16}$O$^{16}$O)$_{HPO\_meas}$, δ($^{16}$O$^{16}$O_/$^{14}$N$^{14}$N)$_{HPO\_meas}$, δ($^{36}$Ar/$^{40}$Ar −)$_{HPO\_meas}$ and δ($^{40}$Ar/$^{14}$N$^{14}$N)$_{HPO\_meas}$ measured by the mass spectrometer. δ(O$_2$/N$_2$)$_{HPO\_meas}$ and δ(Ar/ N$_2$)$_{HPO\_meas}$ calculated using equations (3) and (4), and differences between δ(O$_2$/N$_2$)$_{HPO\_meas}$ and δ($^{16}$O$^{16}$O/$^{14}$N$^{14}$N)$_{HPO\_meas}$ are also shown.

| Cylinder number | δ($^{15}$N$^{14}$N/$^{14}$N$^{14}$N)$_{HPO\_meas}$ | δ($^{17}$O$^{16}$O/$^{16}$O$^{16}$O)$_{HPO\_meas}$ | δ($^{18}$O$^{16}$O/$^{16}$O$^{16}$O)$_{HPO\_meas}$ | δ($^{16}$O$^{16}$O/$^{14}$N$^{14}$N)$_{HPO\_meas}$ | δ(O$_2$/N$_2$)$_{HPO\_meas}$ | δ(O$_2$/N$_2$)$_{HPO\_meas}$ −δ($^{16}$O$^{16}$O/$^{14}$N$^{14}$N) | δ($^{36}$Ar/$^{40}$Ar)$_{HPO\_meas}$ | δ($^{40}$Ar/$^{14}$N$_2$)$_{HPO\_meas}$ | δ(Ar/N$_2$)$_{HPO\_meas}$ |
|---|---|---|---|---|---|---|---|---|---|
| CPC00556 | −2365.0 ± 1.2 | −4032 ± 50 | −7907.8 ± 2.6 | 4477.5 ± 3.2 | 4459.2 ± 3.2 | −18.2 | −2465 ± 50 | 9649.0 ± 6.5 | 9658.1 ± 6.5 |
| CPB28679 | −2343.5 ± 1.2 | −4032 ± 50 | −8298.0 ± 2.6 | −9704.7 ± 3.2 | −9724.4 ± 3.2 | −19.7 | −1969 ± 50 | −14102.6 ± 6.5 | −14092.2 ± 6.5 |
| CPB16178 | −2372.5 ± 1.2 | −4219 ± 50 | −8279.7 ± 2.6 | 12011.7 ± 3.2 | 11991.7 ± 3.2 | −20.0 | −2197 ± 50 | −7828.0 ± 6.5 | −7818.1 ± 6.5 |
| CPB16345 | −2351.5 ± 1.2 | −4676 ± 50 | −9087.6 ± 2.6 | −3624.2 ± 3.2 | −3647.7 ± 3.2 | −23.5 | −2311 ± 50 | 712.0 ± 6.5 | 721.5 ± 6.5 |
| CPB16315 | −2356.2 ± 1.2 | −4665 ± 50 | −9069.6 ± 2.6 | −1946.8 ± 3.2 | −1970.2 ± 3.2 | −23.4 | −2228 ± 50 | −4538.2 ± 6.5 | −4528.5 ± 6.5 |
| CPB16379 | −2416.8 ± 1.2 | −4655 ± 50 | −9062.8 ± 2.6 | −763.6 ± 3.2 | −786.6 ± 3.2 | −22.9 | −2261 ± 50 | −3074.4 ± 6.5 | −3064.3 ± 6.5 |
| CPB16349 | −2407.9 ± 1.2 | −4630 ± 50 | −9036.0 ± 2.6 | 2833.1 ± 3.2 | 2810.2 ± 3.2 | −23.0 | −2360 ± 50 | 1485.7 ± 6.5 | 1495.4 ± 6.5 |
| CPB28912 | −2397.2 ± 1.2 | −4656 ± 50 | −9075.3 ± 2.6 | 554.6 ± 3.2 | 531.5 ± 3.2 | −23.2 | −2348 ± 50 | 1812.2 ± 6.5 | 1821.9 ± 6.5 |
| CPB28679 | −2390.8 ± 1.2 | −5109 ± 50 | −9941.2 ± 2.6 | 212.5 ± 3.2 | 185.4 ± 3.2 | −27.1 | −2338 ± 50 | −642.8 ± 6.5 | −633.2 ± 6.5 |

These values are on the AIST scale, i.e., determined against AIST reference air and are given in per meg.

Numbers following the symbol ± denote the standard uncertainty.

[Figure]

[Figure]

**Figure 5** Relationship between  δ(O$_2$/N$_2$)$_{HPO\_grav}$ and δ(Ar/N$_2$)$_{HPO\_meas}$ on the AIST scale (upper). Fitting residuals  δ(O$_2$/N$_2$)$_{HPO\_meas}$ and δ(Ar/N$_2$)$_{HPO\_meas}$ are likewise shown (lower).

---

## Author Response (AR2)

We wish to express our appreciation for your significant and useful comments. We have revised the manuscript, considering your comments and suggestions.

Reviewer 1: Table 5 column headers are still difficult to read due to automatic line change. Please reformat for better reading.
Response: We revised the column headers.

Reviewer 2: Thanks a lot for your very thorough inspection. I think it is important to study on temperature drift on mass. I would like to kindly request to answer (or describe) below additional points before publication.
1. Please add more information (or reference) on the AIST reference in Page 5.
Response: The sentences were added to describe the additional information, as suggested (P5, L15-19).

2. Please show us the measured weighing values(in mg) of the components in Table 2.
Response: We described rough amount values of filled gases (P4, L5-6). We don't think the measured weighing values is needed for the purpose of the article. We show the measured values in Table A below for reviewer 2. Instead, we added the sentence for the uncertainties of the values because the uncertainties were important (P11, L3-4).

3. Please add uncertainties' sources (how to calculate) of each uncertainty factors as well as each component in Table 3. And please add a summary table (total uncertainty budget of a sample cylinder) with all uncertainty factors.
Response: We attached footnotes (how to calculate) at the bottom of Table 3.

4. Please write (add) their uncertainties of the final purities in Table 4.
Response: Their uncertainties of the final purities were added in Table 4.

5. For decision of $CO_2$ amounts, please describe how to calibrate the CRDS.
Response: We simply described how to calibrate the CRDS (p6, L33-L35).

Table A The measured weighing values.

| Cylinder number | Preparation date | Measured weighing values, mg | | |
|---|---|---|---|---|
| | | $N_2$ | $O_2$ | $CO_2/Ar$ |
| CPC00556 | 15 March 2017 | 854769.3± 1.2 | 262923.0 ± 1.2 | 15438.1 ± 1.2 |
| CPB28679 | 29 March 2017 | 990613.0 ± 1.2 | 300411.9 ± 1.2 | 17468.8 ± 1.2 |
| CPB16178 | 5 April 2017 | 860566.6 ± 1.2 | 266688.2 ± 1.2 | 15243.9 ± 1.2 |
| CPB16345 | 7 April 2017 | 842578.7 ± 1.2 | 257084.5 ± 1.2 | 15052.6 ± 1.2 |
| CPB16315 | 12 April 2017 | 887499.8 ± 1.2 | 271247.5 ± 1.2 | 15770.7 ± 1.2 |
| CPB16379 | 17 April 2017 | 1031180.2 ± 1.2 | 315534.7 ± 1.2 | 18351.7 ± 1.2 |
| CPB16349 | 13 June 2017 | 1013089.7 ± 1.2 | 311111.8 ± 1.2 | 18149.1 ± 1.2 |
| CPB28912 | 15 June 2017 | 1002963.3 ± 1.2 | 307304.7 ± 1.2 | 17975.5 ± 1.2 |
| CPB28679 | 22 June 2017 | 1016867.6 ± 1.2 | 311455.2 ± 1.2 | 18178.9 ± 1.2 |

Numbers following the symbol ± denote the standard uncertainty.